# ROUTE: Robust Multitask Tuning and Collaboration for Text-to-SQL

**Yang Qin**[1] **Chao Chen**[2] **Zhihang Fu**[2] **Ze Chen**[2] **Dezhong Peng**[1,3]* **Peng Hu**[1]* **Jieping Ye**[2]

[1]Sichuan University, [2]Independent Researcher, [3]Tianfu Jincheng Laboratory

## Abstract

Despite the significant advancements in Text-to-SQL (Text2SQL) facilitated by large language models (LLMs), the latest state-of-the-art techniques are still trapped in the in-context learning of closed-source LLMs (*e.g.*, GPT-4), which limits their applicability in open scenarios. To address this challenge, we propose a novel **RO**bust m**U**ltitask **T**uning and collaboration m**E**thod (ROUTE) to improve the comprehensive capabilities of open-source LLMs for Text2SQL, thereby providing a more practical solution. Our approach begins with multi-task supervised fine-tuning (SFT) using various synthetic training data related to SQL generation. Unlike existing SFT-based Text2SQL methods, we introduced several additional SFT tasks, including schema linking, noise correction, and continuation writing. Engaging in a variety of SQL generation tasks enhances the model's understanding of SQL syntax and improves its ability to generate high-quality SQL queries. Additionally, inspired by the collaborative modes of LLM agents, we introduce a Multitask Collaboration Prompting (MCP) strategy. This strategy leverages collaboration across several SQL-related tasks to reduce hallucinations during SQL generation, thereby maximizing the potential of enhancing Text2SQL performance through explicit multitask capabilities. Extensive experiments and in-depth analyses have been performed on eight open-source LLMs and five widely-used benchmarks. The results demonstrate that our proposal outperforms the latest Text2SQL methods and yields promising performance. The code and data are available at here.

## 1 Introduction

Text2SQL has emerged as a popular and practical technology for question answering based on large-scale databases, serving as a crucial link between natural language and database systems (Zhang et al., 2024). Recently, Large Language Models (LLMs) have proven to be an effective solution in Text2SQL (Pourreza & Rafiei, 2024a). Unlike pioneer works (Katsogiannis-Meimarakis & Koutrika, 2021; Xiao et al., 2016; Bogin et al., 2019; Li et al., 2023a;b; Gu et al., 2023), LLM-based methods (Pourreza & Rafiei, 2024a; Gao et al., 2024a; Wang et al., 2023) primarily develop effective prompt engineering techniques, *e.g.*, in-context learning, to motivate LLMs to understand the database schema and generate accurate SQL query. However, accurately aligning entities in natural language questions and databases for SQL generation remains challenging, especially when dealing with complex database schema or semantically complex questions (Li et al., 2024c).

To address these challenging scenarios, recent efforts (Lee et al., 2024; Talaei et al., 2024; Li et al., 2024d) have developed various pipelines that enhance the entire SQL generation process and reduce potential error risks. These improvements include techniques such as Schema Linking (Pourreza & Rafiei, 2024a), Self-correction (Wang et al., 2023), Chain-of-Thought (CoT) (Tai et al., 2023), Reliability Voting (Li et al., 2024d), *etc*. Although these methods have achieved promising results, they often rely on closed-source models (*e.g.*, GPT-4/4o (Achiam et al., 2023)), which can raise potential privacy risks and incur significant overheads when deploying LLMs in practical scenarios. Moreover, while these techniques perform well on GPT-4 or other large-sized LLMs, their effectiveness

---

*Corresponding authors.

may diminish when applied to smaller open-source LLMs (refer to Table 1). This is due to the limited capacity of smaller LLMs to understand complex instructions, showing lower generalizability.

An alternative involves transforming a general LLM into a specialized LLM by injecting Text2SQL-related knowledge through pre-training or SFT (Li et al., 2024b; Yang et al., 2024b; Gu et al., 2023; Roziere et al., 2023). However, most training-based methods only incorporate the SQL generation task in the SFT stage, resulting in a degraded performance in other tasks that are important for Text2SQL capability, such as schema linking. Additionally, training LLMs on a single SQL generation task poses a substantial risk of diminishing performance in understanding instructions, potentially reducing the model's effectiveness in other important SQL-related tasks beyond SQL generation (Appendix A.6). To avoid this dilemma, DTS-SQL separates the schema linking task from Text2SQL and trains specialized LLMs for each to simplify the process (Pourreza & Rafiei, 2024b). However, this approach requires deploying two separate LLMs during the inference stage, which introduces additional overhead and is impractical in real-world applications.

Considering the limitations of existing prompting or training methods, we propose a robust multitask tuning and collaborative prompting framework (**ROUTE**) for Text2SQL generation. The motivation is intuitive: (1) Multitask training not only enhances the model's SQL generation capabilities but also preserves other abilities such as schema linking. (2) Training LLM in various SQL-related tasks, such as schema linking and noise correction, is expected to enhance the model's understanding of SQL syntax. (3) By employing multitask collaborative prompting, the complex Text2SQL task can be decomposed into several simpler sub-tasks, thereby enhancing the accuracy of Text2SQL.

To achieve this goal, in the training stage, we explore a Multitask Supervised Fine-Tuning paradigm (**MSFT**), which endows the LLM with the capabilities in Text2SQL (**TS**), Schema Linking (**SL**), Noise Correction (**NC**) and Continuation Writing (**CW**). In the inference stage, we develop a Multitask Collaboration Prompting (**MCP**) approach to selectively and incrementally generate the SQL queries, potentially reducing the risk of hallucination in complex SQL generation.

Our contributions can be summarized as follows:

- We introduce a multitask SFT training framework, which equips the LLM with a variety of SQL-related specialized capabilities.

- We propose a multitask collaborative prompting strategy that enables the decomposition of the Text2SQL task into several simpler sub-tasks, leveraging specialized capabilities of LLMs.

- We conduct a thorough evaluation of recent open-source LLMs and perform extensive experiments to demonstrate the effectiveness of multitask SFT and to showcase the generalization capabilities of collaborative prompting.

## 2 RELATED WORK

Text-to-SQL (Li et al., 2023b; Gao et al., 2024a; Pourreza & Rafiei, 2024a) aims to understand user intent and convert natural language questions into SQL queries. Existing Text2SQL methods can be roughly categorized into Pre-LLM and LLM-based methods. Pre-LLM approaches mainly exploit the rule modeling (Katsogiannis-Meimarakis & Koutrika, 2021), specialized neural networks (Xiao et al., 2016; Bogin et al., 2019) and pre-trained models (Fu et al., 2023; Li et al., 2023a;b; Gu et al., 2023) to parse and improve SQL generation. The latter has made significant progress recently thanks to LLM's unique emergent abilities in developing Text2SQL solutions, including prompt engineering (Rajkumar et al., 2022; Gao et al., 2024a; Pourreza et al., 2024; Maamari et al., 2024) and LLM fine-tuning/pre-training (Li et al., 2024b; Pourreza & Rafiei, 2024b; Yang et al., 2024b). In this paper, we mainly focus on the LLM-based methods.

**Prompt Engineering.** To unleash the potential of LLMs in the Text2SQL task, a straightforward attempt is to design effective prompting techniques (Liu et al., 2023; Pourreza & Rafiei, 2024a; Gao et al., 2024a; Wang et al., 2023; Lee et al., 2024; Gao et al., 2024b) to guide LLMs. Recent approaches mainly focus on leveraging closed-source models (*e.g.*, ChatGPT and GPT-4) to design innovative instructions or pipelines through the techniques of chain-of-thought (Zhang et al.), LLMs' agents (Wang et al., 2023), question/task decomposition (Pourreza & Rafiei, 2024a), self-debugging (Wang et al., 2023; Chen et al., 2023), schema linking (Li et al., 2024b), few-shot example selection (Gao et al., 2024a), *etc*. For example, DIN-SQL (Pourreza & Rafiei, 2024a) adopts

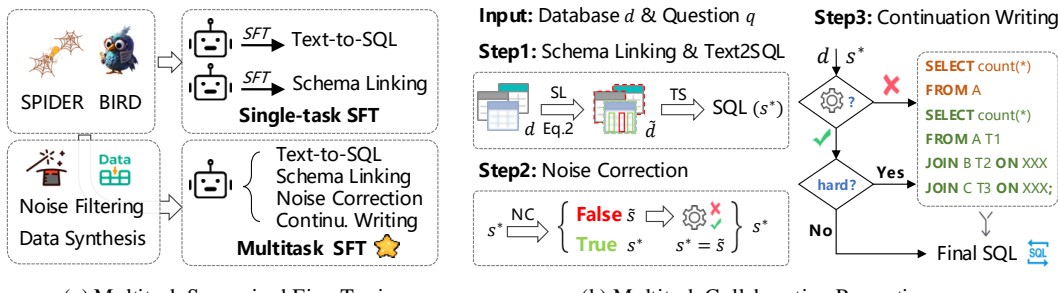

(a) Multitask Supervised Fine-Tuning  (b) Multitask Collaboration Prompting

Figure 1: Overall framework of our ROUTE. Our approach consists of two core stages, *i.e.*, Multitask Supervised Fine-tuning (MSFT) and Multitask Collaboration Prompting (MCP). Different from existing methods that only focus on unitask learning in a single LLM, our MSFT aims to empower LLMs to handle multiple SQL-specific tasks by utilizing synthetic data for supervised fine-tuning. MCP mainly leverages the capabilities of individual tasks for a given database and question to collaboratively generate accurate SQL queries. It enhances the final SQL query incrementally through a three-step process that leverages the multitasking capabilities of LLMs. Note that TS is the abbreviation of Text2SQL, and the task definitions of SL, NC, and CW can be found in Section 3.1.

multiple steps to reduce the complexity of the task for an accurate SQL query. MAC-SQL (Wang et al., 2023) decomposes the original question into several sub-questions and refiner agents to check and correct the final SQL generation. However, these approaches come with huge inference overhead, and may no longer be feasible for small-sized LLMs. In contrast, our method can effectively alleviate these problems through multi-task training and collaborative prompting.

**Fine-tuning LLMs.** Although prompting with closed-source models such as GPT-4 has achieved promising performance, the cost of inference and concerns about data privacy make it particularly urgent to develop specialized LLMs for Text2SQL (Li et al., 2024b; Pourreza & Rafiei, 2024b; Yang et al., 2024b). To break such limitations, CODES (Li et al., 2024b) exploits SQL-related corpus for incremental pre-training with StarCoder (Li et al.) and designs some plug-and-play utils to enhance Text2SQL performance on several benchmarks. DTS-SQL (Pourreza & Rafiei, 2024b) performs supervised fine-tuning (SFT) with a specialized model for schema linking to reduce complexity and improve performance. Recently, SENSE (Yang et al., 2024b) enhanced the preference of LLMs for correct SQL answers by synthesizing strong data and performing Direct Preference Optimization (DPO) (Rafailov et al., 2024) on weak data from weak LLMs. In this paper, we propose to aggregate multiple tasks to enhance the comprehensive SQL-related capabilities and then employ multitask collaboration to minimize potential risks in schema linking errors or SQL clause errors.

## 3 METHODOLOGY

In this section, we will delve into our proposed **RO**bust multitask **T**uning m**E**thod (ROUTE), designed to enhance SQL generation capabilities. As shown in Figure 1, ROUTE consists of two core stages, *i.e.*, Multitask Supervised Fine-Tuning (MSFT) and Multitask Collaboration Prompting (MCP) for SQL Generation. In the following, we first show the notations and definitions used in this work in Section 3.1, then introduce the details of our ROUTE in Sections 3.2 and 3.3 for Text2SQL.

### 3.1 NOTIONS AND PROBLEM FORMULATION

Given an instructional Text2SQL dataset $\mathcal{D} = \{(d_i, q_i, s_i)\}_{i=1}^{N}$, where $d_i$ is a SQL database, $q_i$ is a natural language question that may be accompanied by a question hint, and $s_i$ is a ground-truth SQL query, the purpose of Text2SQL is to exploit an LLM($\mathcal{M}$) to generate a SQL query $s_i^*$ based on a prompt constructed by $d_i$ and $q_i$, and the execution results of predicted SQL query are consistent with those of the ground-truth SQL query $s_i$. In this paper, in addition to the standard Text2SQL task, our approach incorporates three additional SQL-related tasks in the SFT and SQL generation stages, including schema linking, noise correction, and continuation writing, as shown in Figure 2. All these tasks are defined as follows:

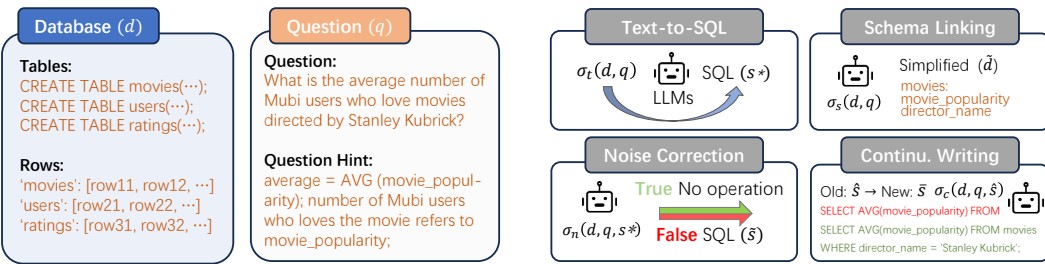

Figure 2: Illustration of all SQL-related tasks involved in our ROUTE. These tasks are based on a given database and user question. We form the prompt for each task by extracting a schema (table) description, a few rows of examples from the database, and the given question (possibly with a hint).

(1) **Text-to-SQL** (Text2SQL, TS) aims to generate a SQL query ($s_i^*$) based on a prompt constructed by the database ($d_i$) and question ($q_i$), and the execution results of $s_i^*$ are consistent with those of its ground-truth SQL query ($s_i$). The function of prompt formatting based on the database and question is represented as $\sigma_t(d_i, q_i)$.

(2) **Schema Linking** (SL) aims to identify relevant tables and columns in the database ($d_i$) for the given question ($q_i$), thus avoiding verbose information in the prompt to reduce complexity, which has been proven to improve Text2SQL performance by recent studies (Pourreza & Rafiei, 2024b;a). The function of prompt formatting is represented as $\sigma_s(d_i, q_i)$.

(3) **Noise Correction** (NC) is required to determine whether the execution results of the predicted SQL query ($s_i^*$) can correctly answer the question ($q_i$) based on the database $d_i$. If not, LLM ($\mathcal{M}$) will be asked to provide a revised SQL query ($\tilde{s}_i$). The function of prompt formatting is represented as $\sigma_n(d_i, q_i, s_i^*)$.

(4) **Continuation Writing** (CW) is another strategy to refine SQL. Given an incomplete SQL query ($\hat{s}_i$), LLM ($\mathcal{M}$) is required to continue writing it into a complete and valid SQL query ($\bar{s}_i$) whose execution results can correctly answer the question ($q_i$). Likewise, the function of prompt formatting can be represented as $\sigma_c(d_i, q_i, \hat{s}_i)$.

From the task definitions provided, it is evident that the tasks of TS and NC have the potential to directly enhance the quality of the SQL queries, as they are directly related to SQL generations. Empirical evidence from previous works (Pourreza & Rafiei, 2024a; Lee et al., 2024) has also demonstrated that SL can significantly reduce prompt complexity by simplifying database information, thereby improving performance. In addition, we define a new task named Continuation Writing (CW), which indirectly contributes to the final SQL generation. The motivation of Continuation Writing (CW) stems from the fact that LLMs possess inherent continuation capabilities, which make it easier than requiring the model to generate complete SQL queries. To unleash the potential of these tasks, in this paper, we first enhance the specialized capabilities of LLMs by aggregating multiple tasks for supervised fine-tuning. Then, we exploit the collaboration of multiple tasks to reduce potential risks in schema linking errors or SQL clause errors during SQL generation, thus further improving performance. Note that the prompt templates shown in Figure 2 for all the above tasks can be found in Appendix A.11.

## 3.2 MULTITASK SUPERVISED FINE-TUNING

Existing SFT-based methods (Li et al., 2024b; Yang et al., 2024b) mainly focus on the task of Text2SQL to improve final performance. However, previous prompting-based methods have shown that SQL-related tasks can also effectively improve performance. Unfortunately, due to the lack of specialized pre-training or instruction fine-tuning for these related tasks, it is difficult for open-source LLMs to complete these tasks with high accuracy. In this section, we present the details of our Multitask Supervised Fine-tuning (MSFT) on the dataset $\mathcal{D}$ that combines training sets from two wide-used cross-domain datasets, *i.e.*, SPIDER (Yu et al., 2018) and BIRD (Li et al., 2024c).

**Noisy correspondence filtering.** Recent empirical evidence (Wretblad et al., 2024; Wretblad & Gordh Riseby, 2024) indicates that even carefully annotated Text2SQL datasets exhibit semantic inconsistencies between the given question and the ground-truth SQL query, as shown in Figure 3, which we call noisy correspondences and widely exist in various field (Qin et al., 2022; 2023; Sun

et al., 2024; Qin et al., 2024). It is widely recognized that noise remains a primary factor contributing to hallucinations in existing LLMs. To mitigate this, we fine-tuned the selected LLM (Llama3-8B) to serve as a noise discriminator, specifically designed to detect potential noise in the corpus. To construct corresponding SFT data, we construct a positive discriminant example $\langle \sigma_n(d_i, q_i, s_i), A_{\text{pos}} \rangle$ and a negative discriminant example $\langle \sigma_n(d_i, q_i, s_i'), A_{\text{neg}}(s_i) \rangle$ for each Text-SQL pair, where $s_i$ is the ground-truth SQL query and $A_{\text{pos}}/A_{\text{neg}}$ means the affirmative/negative answer, *i.e.*, $A_{\text{pos}}$: 'The execution results of the SQL query can correctly answer the question.'; $A_{\text{neg}}(s_i)$: 'The execution results of the SQL query cannot correctly answer the question. The correct SQL query should be: $\{s_i\}$'.

To obtain negative SQL examples for the construction of $A_{\text{neg}}$, we employ open-source LLMs Qwen2-7B and Llama3-8B to generate all SQL responses for the questions in $\mathcal{D}$ in a zero-shot manner and then compare the execution results of each predicted SQL query with those of the ground-truth SQL query. If they do not match, we view the predicted SQL query as a negative example ($s_i'$). The goal is consistent with the defined task of NC shown in Section 3.1. In addition, to enrich the diversity of SFT data, we artificially and randomly introduce five types of errors in ground-truth SQL queries, including schema linking errors, nesting errors, GROUP BY errors, JOIN errors, and

> **Example 1:**
> **Q1:** What is the number of inhabitants and income of geographic identifier 239?
> **A1:** SELECT INHABITANTS_K FROM Demog WHERE GEOID = 239;
> **R1:** SELECT INHABITANTS_K, INCOME_K FROM Demog WHERE GEOID = 239;
>
> **Example 2:**
> Q2: List the geographic id of places where the income is above average.
> **A2:** SELECT AVG(INCOME_K) FROM Demog;
> **R2:** SELECT GEOID FROM Demog WHERE INCOME_K > ( SELECT AVG(INCOME_K) FROM Demog );

Figure 3: The examples of noisy pairs in the BIRD training set. R1 and R2 are the corrected SQL queries. More noisy examples can be found in Appendix A.12.

symbol errors. See Appendix A.7 for more details. Then, we fine-tune a Llama3-8B model through a standard SFT process, and finally, we perform inference on $\mathcal{D}$ to identify and filter noise, achieving self-purification of the training data set. For convenience, we represent the purified data as $\tilde{\mathcal{D}}$.

**Synthesizing data for MSFT.** Our MSFT consists of one major task (Text2SQL) and three minor tasks defined in Section 3.1, namely SL, NC, and CW, to empower LLMs with specialized capabilities. To this end, we need to synthesize or construct SFT datasets from $\tilde{\mathcal{D}}$ for each specific task.

(1) For the SFT data of Text2SQL task, we utilize the defined function $\sigma_t$ to construct the `Prompt` and the ground-truth SQL query serves the `Response`. The SFT data constructed for Text2SQL is denoted as $\mathcal{D}_t = \{\sigma_t(d_i, q_i), s_i\}_{i=1}^{N_t}$, where $N_t$ is the corresponding data size.

(2) For the SFT data of schema linking task, we utilize the ground-truth SQL query to exact the tables and columns of the corresponding database as the `Response`, which is denoted as $f(s_i, d_i)$ and $f$ is the parsing function. If there is a `COUNT(*)` statement, we only retain the primary keys of the corresponding table. We represent the dataset as $\mathcal{D}_s = \{\sigma_t(d_i, q_i), f(s_i, d_i)\}_{i=1}^{N_s}$.

(3) For the SFT data of noise correction task, the synthesis and construction of the data are consistent with that of the dedicated discriminator mentioned above, including some positive examples and some corresponding negative examples. For convenience, we unify the expression as $\mathcal{D}_n = \{\sigma_n(d_i, q_i, s_i), A_{\text{pos}}\}_{i=1}^{N_p} \cup \{\sigma_n(d_i, q_i, s_i'), A_{\text{neg}}(s_i)\}_{i=1}^{N_n}$, where $N_p$ and $N_n$ are the numbers of positive and negative examples, respectively.

(4) For the SFT data of continuation writing task, we construct the `Prompt` by truncating the ground-truth SQL query from a random position, and the corresponding ground-truth SQL query is used as the `Response`. Likewise, the SFT data constructed for CW is denoted as $\mathcal{D}_c = \{\sigma_t(d_i, q_i, \hat{s}_i), s_i\}_{i=1}^{N_c}$, where $N_c$ is the corresponding data size and $\hat{s}_i$ is the truncated SQL query.

Currently, our multitask supervised fine-tuning (MSFT) dataset is defined as $\mathcal{D}_M = \mathcal{D}_t \cup \mathcal{D}_s \cup \mathcal{D}_n \cup \mathcal{D}_c$. We will then proceed to perform SFT to enable LLMs to handle these tasks explicitly.

**Fine-tuning LLMs through MSFT.** Given above MSFT data $\mathcal{D}_M$, wherein the input prompt and target response generated are represented $\boldsymbol{x}$ and $\boldsymbol{y}$ for convenience, the supervised fine-tuning for specialized LLMs can be formulated as maximizing the log-likelihood objective:

$$\mathbb{E}_{(\boldsymbol{x}, \boldsymbol{y}) \sim \mathcal{D}_M} \left[ \sum_{t=1}^{T} \log p_{\mathcal{M}}(y_t | \boldsymbol{y}_{1:t-1}, \boldsymbol{x}) \right], \quad (1)$$

where $T$ is the sequence length of $\boldsymbol{y}$. After completing the MSFT stage, we can obtain a specialized LLM that is capable of handling various SQL-related tasks, which allows us to perform multitask collaboration to improve final SQL generation.

## 3.3 MULTITASK COLLABORATION PROMPTING

To fully utilize these specialized capabilities of LLMs, we develop a Multitask Collaboration Prompting (MCP) method to reduce potential risks in schema linking errors or SQL clause errors in SQL generation as shown in Figure 1-(b). Our MCP consists of three core steps corresponding to the defined task in Section 3.1.

First, to streamline the redundant information in the prompt, we propose an enhanced schema linking strategy, which leverages the schema linking capability of LLMs to identify tables and columns relevant to solving the user question, complemented by the pseudo-SQL query (Li et al., 2024d) to simplify the database. Given a database $d_i$ and a user question $q_i$, the pseudo-SQL refers to the intermediate SQL query generated in advance using the complete schema information, *i.e.*, $\mathcal{M}(\sigma_t(d_i, q_i), d_i)$. The final simplified database $\tilde{d}_i$ can be represented as:

$$\tilde{d}_i = \mathcal{M}(\sigma_s(d_i, q_i)) \uplus f(\mathcal{M}(\sigma_t(d_i, q_i), d_i), \tag{2}$$

where $f$ is the parsing function to extract the tables and columns from SQL queries through fuzzy matching and $\uplus$ denotes the operation defined for merging tables and columns. Such a merger operation helps to decrease the likelihood of missing potentially related entities (tables or columns) and maximize the information relevant to the question, thereby reducing the potential risks in schema linking errors. After obtaining the tables and columns by schema linking, we conduct SQL generation by $s_i^* = \mathcal{M}(\sigma_t(\tilde{d}_i, q_i))$ to obtain an intermediate SQL query. Then, to explicitly identify the incorrect SQL queries, we utilize LLMs to check them by combing $\sigma_n(d_i, q_i, s^*)$ with the exception information $e_i$ raised by a SQL executor $(\text{SQLer}(d_i, s^*))$, *i.e.*, $\mathcal{M}(\sigma_n(d_i, q_i, s^*, e_i))$. Note that $e_i$ is empty if the SQL query is executed successfully. If the LLM shows that $s_i^*$ cannot answer $q_i$ accurately, we take the corrected SQL $\tilde{s}_i$ and check $\tilde{s}_i$ using the SQL executor $(\text{SQLer}(d_i, \tilde{s}_i))$. If it executes successfully, replace $s^*$ with $\tilde{s}_i$.

Finally, we present a novel strategy stemming from our observations of LLMs on SQL queries after continuation writing. We discovered that continuation writing of a given SQL piece by LLMs can enhance the quality of the generated SQL. In view of this finding, we employ LLMs to continue writing truncated SQL queries to refine and improve complex SQL queries. To achieve this, we categorize the difficulty of the generated SQL query as follows: *Simple*: involving only one table, *Medium*: involving two tables, and *Hard*: involving more than two tables. For ease of presentation, we define a difficulty evaluation function $h(s_i, d_i) \in \{1, 2, 3\}$, where $1 \sim 3$ correspond to three hardness levels. For all *Hard* SQL queries, we conduct continuation writing on the incomplete SQL queries started with '`SELECT`' for further refinement. To enhance the clarity of our MCP, we offer a detailed explanation of the process in Appendix A.9.

## 4 EXPERIMENTS

### 4.1 EXPERIMENT SETTINGS

**Benchmarks.** To evaluate our method, we conduct extensive experiments on five benchmarks to verify the effectiveness of our method. These benchmark includes two widely-used cross-domain benchmarks, *i.e.*, SPIDER (Yu et al., 2018) and BIRD (Li et al., 2024c), and three robust benchmarks derived from SPIDER, *i.e.*, SPIDER-SYN (Gan et al., 2021a), SPIDER-DK (Gan et al., 2021b), and SPIDER-Realistic (Deng et al., 2020). SPIDER consists of 7,000 Text-SQL pairs in the training set, 1,034 pairs in the development set, and 2,147 pairs in the test set, which covers nearly 200 databases and 138 domains. BIRD is a recently proposed benchmark including 9,428, 1,534, and 1,789 pairs in training, development, and test sets, respectively. Compared with SPIDER, BIRD contains more complex databases, more difficult questions, and external knowledge, making it more challenging. For the derived variants, SPIDER-SYN replaces some keywords in the questions in the SPIDER dev set with synonyms. SPIDER-DK introduces some domain knowledge reasoning challenges, while SPIDER-Realistic removes explicit mentions of column names in the SPIDER development set. These variants all simulate real-world scenarios for a more comprehensive evaluation.

Table 1: Performance comparison on SPIDER and BIRD benchmarks. The results of re-evaluation using the open-source code repository are marked with '†'. In groups of open-source LLMs, the best results are highlighted in **bold** and the second-best results are in underlined.

| Methods | SPIDER | | | BIRD | |
|---|---|---|---|---|---|
| | Dev-EX | Dev-TS | Test-EX | Dev-EX | Dev-VES |
| *Prompting with GPT-4* | | | | | |
| GPT-4 (Achiam et al., 2023) | 72.9 | 64.9 | - | 46.4 | 49.8 |
| DIN-SQL + GPT-4 (Pourreza & Rafiei, 2024a) | 82.8 | 74.2 | 85.3 | 50.7 | 58.8 |
| DAIL-SQL + GPT-4 (Gao et al., 2024a) | 83.5 | 76.2 | 86.6 | 54.8 | 56.1 |
| MAC-SQL + GPT-4 (Wang et al., 2023) | 86.8 | - | 82.8 | 59.4 | 66.2 |
| MCS-SQL + GPT-4 (Lee et al., 2024) | 89.5 | - | 89.6 | 63.4 | 64.8 |
| *Prompting with Open-Source LLMs* | | | | | |
| Mistral-7b (Jiang et al., 2023) | 56.8 | 47.3 | 60.1 | 22.5 | 27.8 |
| Llama3-8B (Touvron et al., 2023) | 69.3 | 58.4 | 69.1 | 32.1 | 31.6 |
| Qwen2.5-7B (Yang et al., 2024a) | 72.5 | 64.0 | 75.9 | 41.1 | 42.0 |
| Qwen2.5-14B (Yang et al., 2024a) | 76.9 | 66.3 | 78.4 | 48.4 | 49.2 |
| DIN-SQL + Llama3-8B | 48.7 | 39.3 | 47.4 | 20.4 | 24.6 |
| DIN-SQL + Qwen2.5-7B | 72.1 | 61.2 | 71.1 | 30.1 | 32.4 |
| MAC-SQL + Llama3-8B | 64.3 | 52.8 | 65.2 | 40.7 | 40.8 |
| MAC-SQL + Qwen2.5-7B | 71.7 | 61.9 | 72.9 | 46.7 | 49.8 |
| **Ours**: MCP + Llama3-8B | 75.0 | 63.4 | 72.0 | 42.7 | 44.8 |
| **Ours**: MCP + Qwen2.5-7B | 78.3 | 67.2 | 78.7 | 49.7 | 52.8 |
| **Ours**: MCP + Qwen2.5-14B | **80.0** | **67.3** | **80.6** | **56.3** | **57.6** |
| *Fine-Tuning with Open-Source LLMs* | | | | | |
| Llama3-8B + SFT (Touvron et al., 2023) | 82.4 | 76.2 | 83.1 | 53.1 | 59.0 |
| Qwen2.5-7B + SFT (Yang et al., 2024a) | 80.9 | 75.6 | 82.8 | 51.4 | 53.1 |
| DTS-SQL-7B (Pourreza & Rafiei, 2024b) | 82.7† | 78.4† | 82.8† | 55.8 | 60.3 |
| CODES-7B + SFT (Li et al., 2024b) | 85.4 | 80.3 | - | 57.2 | 58.8 |
| CODES-15B + SFT (Li et al., 2024b) | 84.9 | 79.4 | - | 58.5 | 56.7 |
| SENSE-7B (Yang et al., 2024b) | 83.2 | 81.7 | 83.5 | 51.8 | - |
| SENSE-13B (Yang et al., 2024b) | 84.1 | **83.5** | 86.6 | 55.5 | - |
| **Ours**: ROUTE + Llama3-8B | 86.0 | 80.3 | 83.9 | 57.3 | 60.1 |
| **Ours**: ROUTE + Qwen2.5-7B | 83.6 | 77.5 | 83.7 | 55.9 | 57.4 |
| **Ours**: ROUTE + Qwen2.5-14B | **87.3** | 80.9 | **87.1** | **60.9** | **65.2** |

**Evaluation Metrics.** In our experiments, following previous works (Li et al., 2024b; Yang et al., 2024b), we use the execution accuracy (EX) and test-suite accuracy (TS) to evaluate the performance of Text2SQL on SPIDER and its variant benchmarks. More specifically, except for being unable to report TS on the SPIDER-DK and SPIDER test set, we report EX and TS on all others. For BIRD, following its official settings, we report EX and an indicator called Valid Efficiency Score(VES) that considers execution efficiency to evaluate performance. Note that for all metrics, higher is better.

**Implementation Details.** We choose the popular LLMs Llama3-8B-Instruct (Touvron et al., 2023) and Qwen2.5-7B/14B-Instruct (Team, 2024) as our investigated models. We use the Llama-Factory framework (Zheng et al., 2024) to conduct SFT and MSFT for reproducibility. We conduct experiments on 8×A100 GPUs with a batch size of 64 (32 for 14B-sized LLM). LLMs are fine-tuned for two epochs using AdamW with the learning rate of $1e$-5 that decayed to $0$ at the end of training by a cosine scheduler. During inference, the temperature is set to $0.01$ to ensure reproducibility.

**Compared Baselines.** Our baselines can be categorized into three groups, *i.e.*, the prompting methods with GPT-4, the prompting methods with open-source LLMs, and fine-tuning-based methods with open-source LLMs. The first group includes DIN-SQL (Pourreza & Rafiei, 2024a), MAC-SQL (Wang et al., 2023), DAIL-SQL (Gao et al., 2024a), MCS-SQL (Lee et al., 2024), and the corresponding closed-source LLM GPT-4 (Achiam et al., 2023). For the baselines with open-source LLMs, we choose some popular LLMs and report the zero-shot performance. Besides, we also apply the prompting method, *i.e.*, DIN-SQL, and MAC-SQL, to Llama3 (Touvron et al., 2023) and Qwen2.5 (Team, 2024) to explore the robustness. For the last group, it is represented by specialized LLMs, including DTS-SQL (Pourreza & Rafiei, 2024a), CODES (Li et al., 2024b), SENSE (Yang et al., 2024b), and the base LLMs fine-tuned on the SPIDER and BIRD training sets. To be fair, we re-evaluate the performance of some of the baselines using the open-source code repositories.

## 4.2 Comparison Results

**Results on SPIDER and BIRD.** In Table 1, we report the performance of our method and baselines on the SPIDER development set, test set, and BIRD development set. Due to time limitations, we are unable to provide the results of our ROUTE and MCP on the BIRD test set. From the results, we can see that the prompting-based baselines achieve promising performance with the help of GPT-4, however, their effectiveness is obviously limited in the small-sized open-source LLMs. In contrast, our MCP effectively improves the performance of these small-sized LLMs, *e.g.*, MCP brings more than 5% absolute performance improvement to Qwen2.5-7B on the SPIDER development set. For the fine-tuning group, our ROUTE has achieved the best results in most metrics among the fine-tuning-based methods. Especially on the BIRD dataset, our method (14B) surpasses some existing prompting-based methods (*e.g.*, DIN-SQL, DAIL-SQL, and MAC-SQL) with an EX score of **60.8**, greatly narrowing the gap with existing methods using GPT-4.

**Results on SPIDER-variants.** Table 5 shows the performance on the benchmarks derived from SPIDER. Except for SPIDER-SYN, our MSFT can slightly improve performance on other variant benchmarks. Despite this, our ROUTE can perform better by inspiring the multi-task capabilities of LLMs with the help of MCP, thus improving performance. In addition, MCP can still have performance gains for the SFT model, demonstrating the necessity of conducting multitask collaboration.

Table 2: Performance on SPIDER-variant benchmarks. The best results are highlighted in **bold**.

| Methods | SYN | | Realistic | | DK | Avg. |
| | EX | TS | EX | TS | EX | |
|---|---|---|---|---|---|---|
| Llama3-8B | 60.3 | 47.1 | 68.5 | 50.8 | 58.3 | 57.0 |
| + SFT | 75.3 | 68.7 | 76.8 | 69.7 | 72.0 | 72.5 |
| + SFT + MCP | 76.1 | 69.4 | 78.0 | 70.7 | 73.5 | 73.5 |
| + MSFT | 72.1 | 65.1 | 77.0 | 68.1 | 72.3 | 70.9 |
| + **ROUTE** | **77.4** | **70.2** | **80.9** | **72.6** | **74.6** | **75.1** |

## 4.3 Ablation and Analytic Studies

In this section, we first conduct a comprehensive ablation study on all benchmarks to explore the impact of each key component, verifying the effectiveness of our method. Furthermore, we conduct in-depth analyses to explore the transferability and upper-bound performance of our approach. If not stated, all results are performed on Llama3-8B-Instruct and evaluated on the SPIDER and BIRD development sets using the EX score.

Table 3: The ablation results (EX) on SPIDER and BIRD development sets.

| No. | SFT | MSFT | MCP | NF | SPIDER | BIRD |
|---|---|---|---|---|---|---|
| #1 | ✓ | ✓ | ✓ | ✓ | 86.0 | 57.3 |
| #2 | ✓ | ✓ | | ✓ | 83.6 | 53.6 |
| #3 | ✓ | ✓ | ✓ | | 84.5 | 57.4 |
| #4 | ✓ | ✓ | | | 83.3 | 53.1 |
| #5 | ✓ | | | | 82.4 | 53.1 |
| #6 | ✓ | | ✓ | ✓ | 83.5 | 56.1 |
| #7 | ✓ | | | ✓ | 83.1 | 52.9 |
| #8 | ✓ | | ✓ | | 83.8 | 56.0 |
| #9 | | | ✓ | | 75.0 | 42.7 |
| #10 | | | | | 69.3 | 32.1 |

Table 4: The ablation results (EX) on multi-task collaboration prompting.

| No. | SL | NC | CW | SPIDER | BIRD |
|---|---|---|---|---|---|
| #1 | ✓ | ✓ | ✓ | 86.0 | 57.3 |
| #2 | ✓ | | | 85.8 | 56.0 |
| #3 | | ✓ | | 83.9 | 54.7 |
| #4 | | | ✓ | 83.8 | 54.0 |
| #5 | | | | 83.6 | 53.6 |
| #6 | ✓ | ✓ | ✓ | 75.0 | 42.7 |
| #7 | ✓ | | | 73.3 | 36.8 |
| #8 | | ✓ | | 72.1 | 38.1 |
| #9 | | | ✓ | 71.3 | 36.4 |
| #10 | | | | 69.3 | 32.1 |

**Study on ROUTE.** Table 3 reports the results of ablation experiments of our ROUTE, where '✓' means that the item is adopted and 'NF' means noisy correspondence filtering introduced in Section 3.2. From the results, we have the following observations. First, our MCP can effectively improve the performance of LLMs before and after MSFT, *i.e.*, #10: 69.3/32.1 to #9: 75.0/42.7 and #2: 83.6/53.6 to #1: 86.0/57.3, which demonstrates its effectiveness. Second, applying noisy correspondence filtering significantly improves the performance on SPIDER, *i.e.*, 84.5 *vs.* 86.0, but slightly degrades the performance on BIRD, which indicates that the hard/noisy samples in BIRD hinder the LLM from learning the correct patterns for easy samples. This also suggests that reducing the noise degree and balancing the hard and easy examples in the SFT data is still important for the

LLM to learn the correct SQL-related knowledge. However, in general, the full version of ROUTE achieved the best performance, verifying the importance and effectiveness of each design.

**Study on MCP.** In our ROUTE, MCP can stimulate and give full play to the advantages of multitasking collaboration for accurate SQL generation. To this end, we conduct detailed ablation experiments on MCP to explore the impact of each task. The experimental results are presented in Table 4, where #1~#5 are for the LLM after MSFT and #6~#10 are for the base LLM. We can see that although SL plays a dominant role in performance improvement, it still cannot achieve optimal performance without the full MCP, which shows the complementarity of multiple tasks.

**Study on Enhanced Schema Linking.** As mentioned above, SL plays a leading role in the performance improvement of MCP, which is attributed to the fact that our SL adopts a fusion enhancement strategy, *i.e.*, the combination of the predicted schema linking of LLMs ($SL_{\sigma_s}$) and the extracted entities from a pseudo-SQL query ($SL_{\sigma_t}$). The experimental results are shown in Table 5, where the first four rows are the results of Llama3-8B after MSFT, and the last four rows are the results of the base model. From the results, $SL_{\sigma_t}$ is better than $SL_{\sigma_s}$, as proven by Li et al. (2024d). However, $SL_{\sigma_t}$ can easily discard some seemingly irrelevant but important database entities, thereby generating incorrect SQL queries for degraded performance (*e.g.*, 69.3 to 64.5 on SPIDER development set). Therefore, combining $SL_{\sigma_s}$ and $SL_{\sigma_t}$ can effectively alleviate the issue and achieve a better and more promising performance. In addition, to further understand the performance of our schema linking module, we provide more results and discussions in Appendix A.4.

**Study on Transferability.** To explore the transferability of our ROUTE across different LLMs and different model sizes, we selected several open-source LLMs for experiments, *i.e.*, General LLMs: Llama3-8B/70B (Touvron et al., 2023), Qwen2-7B (Yang et al., 2024a), Qwen2.5-7B/72B (Team, 2024); Code Specialized LLMs: CodeLlama-7B (Roziere et al., 2023), Deepseek-Coder-7B (Guo et al., 2024), and Qwen2.5-Coder-7B (Team, 2024). Due to training costs, we only considered MSFT on the models with a size of around 7B. To explore the transferability of our method on LLMs with different model sizes, we only apply MCP to them for Text2SQL. From Figure 4, one can see that performing SFT on a single task only using Text2SQL data will cause the model to lose the ability of other tasks, resulting in poor MCP performance, while MSFT

Table 5: The ablation results (EX) on enhanced schema linking.

| $SL_{\sigma_s}$ | $SL_{\sigma_t}$ | SPIDER | BIRD |
|:---:|:---:|:---:|:---:|
| ✓ | ✓ | 85.8 | 56.0 |
| ✓ |   | 84.1 | 52.7 |
|   | ✓ | 85.0 | 54.5 |
|   |   | 83.3 | 53.1 |
| ✓ | ✓ | 73.3 | 36.8 |
| ✓ |   | 64.5 | 30.4 |
|   | ✓ | 73.1 | 35.4 |
|   |   | 69.3 | 32.1 |

will retain these capabilities for multitask collaboration, thereby generating more accurate SQL queries. Even for Qwen2.5-Coder which is specially considered for SPIDER and BIRD, our scheme still brings performance improvement. From Table 6, we can see that our MCP is not only applicable to small-sized LLMs but can also improve the Text2SQL performance of LLMs with sizes of around 70B, which shows its promising transferability on different model sizes. Especially for the challenging dataset BIRD, the absolute improvement empowered by our MCP exceeds **6.5%** on average. This shows that our method has good transferability and generalization.

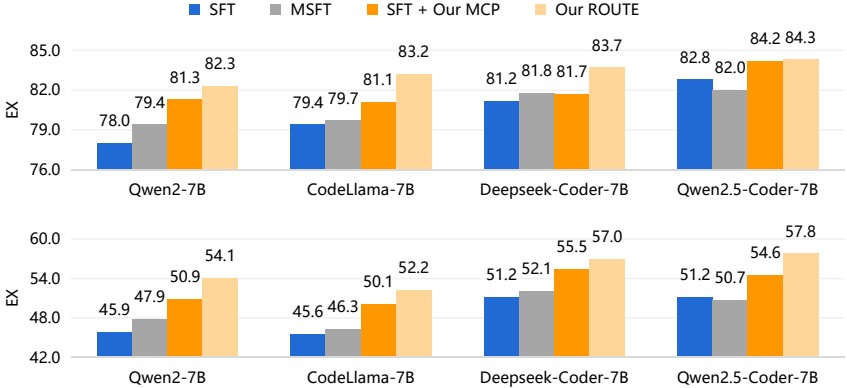

Figure 4: The transferability results on different open-source LLMs on SPIDER (the first row) and BIRD (the second row). See Appendix A.2 for detailed results.

**Study on Upper-bound Performance.** In this section, we explore the upper bound performance of our MCP and ROUTE to understand the potential of multitask collaboration. As shown in Table 7, we define two upper bounds: the first is to use a ground truth simplified database to obtain an ideal schema linking, denoted as U1. The second is to use half of the ground truth SQL query as a hint to refine the wrong or challenging SQL by continuation writing, denoted as U2. From the results, ideal schema linking can effectively achieve accurate SQL generation, which shows that schema linking is still an effective means to solve Text2SQL at this stage and in the future. Besides, U2 can also significantly improve performance, which shows that continuation writing has the potential to become another high-precision solution for Text2SQL. By combining them, our MCP and ROUTE show amazing performance on LLMs only with around 7B size.

Table 7: The upper-bound performance (EX).

Table 6: The performance (EX) of various-sized open-source LLMs.

| Methods | MCP | | ROUTE | |
|---|---|---|---|---|
| | SPIDER | BIRD | SPIDER | BIRD |
| Llama3-8B | 75.0 | 42.7 | 86.0 | 57.3 |
| + U1 | 79.6 | 49.8 | 87.4 | 69.6 |
| + U2 | 76.8 | 47.5 | 87.3 | 61.2 |
| + U1 + U2 | 80.2 | 53.3 | 88.8 | 64.2 |
| Qwen2.5-7B | 78.3 | 49.7 | 83.6 | 55.9 |
| + U1 | 82.2 | 58.9 | 87.3 | 61.5 |
| + U2 | 78.7 | 54.7 | 85.2 | 60.8 |
| + U1 + U2 | 83.0 | 60.4 | 88.2 | 64.9 |

| Models | ≈7B | | ≈70B | |
|---|---|---|---|---|
| | SPIDER | BIRD | SPIDER | BIRD |
| Llama3 | 69.3 | 32.1 | 77.9 | 46.9 |
| Llama3 + MCP | 75.0 | 42.7 | 79.0 | 51.8 |
| Qwen2.5 | 72.5 | 41.1 | 81.7 | 53.3 |
| Qwen2.5 + MCP | 78.3 | 49.7 | 82.3 | 57.1 |

## 5 LIMITATIONS AND BROADER IMPACTS

Although our exploration has achieved promising performance, we have to acknowledge the following limitations. First, our solution may bring additional reasoning costs. Although some existing efficient inference frameworks can alleviate this (*e.g.*, VLLM (Kwon et al., 2023)), we still encourage the exploration of more efficient multitask collaboration modes. Second, from the study on the upper bound performance, there is still a large gap between the performance of our method and the upper bound ones, which encourages further study on synthesis data and SFT paradigms of SQL-related tasks to understand and mitigate the biases and risks potentially brought by multitask collaboration. Finally, we provide the ethics and reproducibility statement in Appendix A.13 to avoid the potential impact and risk.

## 6 CONCLUSION

In this paper, we study and propose a robust multitask tuning and collaboration method named ROUTE to stimulate the potential of open-source LLM in Text2SQL, narrowing the gap with existing solutions based on closed-source LLMs, such as GPT-4. Our method minimizes the risk of hallucination in SQL generation by explicitly learning multiple SQL-related tasks and conducting multitask collaboration. We apply our approach to recent LLMs to demonstrate its effectiveness and superiority on multiple benchmarks. The results show that our method has satisfactory transferability and achieves promising execution accuracy on Text2SQL. In the future, we plan to explore more SQL-relevant tasks, larger LLMs, and more efficient collaboration frameworks for robust Text2SQL.

ACKNOWLEDGMENTS

This work was supported in part by the National Key R&D Program of China under Grant 2024YFB4710604; in part by NSFC under Grant 62472295, 62372315, 62176171 and U21B2040; in part by Sichuan Science and Technology Planning Project under Grant 2024NSFTD0047 and 2024NSFTD0038; in part by System of Systems and Artificial Intelligence Laboratory pioneer fund grant; in part by the Fundamental Research Funds for the Central Universities under Grant CJ202303 and CJ202403; in part by Chengdu Science and Technology Project (2023-XT00-00004-GX); in part by the Sichuan Science and Technology Program (2024NSFTD0049, 2024YFHZ0089).

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

# A    APPENDIX

## A.1    PERFORMANCE COMPARISON WITH DIFFERENT HARDNESS

To explore a more fine-grained performance comparison, we follow previous works (Pourreza & Rafiei, 2024a; Gao et al., 2024a) and report the EX scores on development sets of SPIDER and BIRD. From the results shown in Tables 8 and 9, the conclusion of our ROUTE's overall performance is consistent with that of its fine-grained performance, which verifies the superiority of our method.

Table 8: The performance (EX) comparison with different hardness on the SPIDER development set. The best results in the fine-tuning group are highlighted in **bold**.

| Methods | Easy | Medium | Hard | Extra | All |
|---|---|---|---|---|---|
| *Prompting-based methods* | | | | | |
| C3-SQL + ChatGPT (Dong et al., 2023) | 92.7 | 85.2 | 77.6 | 62.0 | 82.0 |
| DIN-SQL + GPT-4 (Pourreza & Rafiei, 2024a) | 92.3 | 87.4 | 76.4 | 62.7 | 82.8 |
| DAIL-SQL + GPT-4 (Gao et al., 2024a) | 91.5 | 89.2 | 77.0 | 60.2 | 83.1 |
| DAIL-SQL + GPT-4 + Self-consistency | 91.5 | 90.1 | 75.3 | 62.7 | 83.6 |
| SuperSQL + GPT-4 (Li et al., 2024a) | 94.4 | 91.3 | 83.3 | 68.7 | 87.0 |
| MCS-SQL + GPT-4 (Lee et al., 2024) | 94.0 | 93.5 | 88.5 | 72.9 | 89.5 |
| *Fine-tuning-based methods* | | | | | |
| Graphix-3B + PICARD (Li et al., 2023b) | 92.3 | 86.3 | 73.6 | 57.2 | 80.9 |
| RESDSQL-3B (Li et al., 2023a) | 94.8 | 97.7 | 73.0 | 56.0 | 81.8 |
| DTS-SQL (Pourreza & Rafiei, 2024b) | 92.7 | 90.1 | 74.1 | 56.6 | 82.7 |
| RESDSQL-3B + NatSQL (Li et al., 2023a) | 94.4 | 87.9 | 77.0 | 66.3 | 84.1 |
| CODES-7B + SFT (Li et al., 2024b) | 94.8 | 91.0 | 75.3 | 66.9 | 85.4 |
| CODES-15B + SFT (Li et al., 2024b) | 95.6 | 90.4 | 78.2 | 61.4 | 84.9 |
| **Ours:** ROUTE + Llama3-8B | **96.0** | **93.0** | 75.3 | 63.3 | 86.0 |
| **Ours:** ROUTE + Qwen2.5-7B | 92.7 | 89.7 | 77.0 | 60.2 | 83.6 |
| **Ours:** ROUTE + Qwen2.5-14B | 94.0 | **93.0** | **81.6** | **68.1** | **87.3** |

## A.2    THE RESULTS STUDIED ON TRANSFERABILITY

In this appendix, we provide detailed results of a transferability study in Table 10. Furthermore, we compare our methods with those using the same base LLMs, *i.e.*, SQL-Llama7B (Wang et al., 2023), DTS-SQL (Pourreza & Rafiei, 2024b) and SENSE (Yang et al., 2024b). The results show that our method has good scalability on various LLMs, whether the base or code-specific ones. This is sufficient to verify the superiority and generalization of our ROUTE.

## A.3    MORE COMPARISONS WITH RECENT WORKS

In this appendix, we provide more comparisons with recent works in Appendix A.3, including CHASE-SQL (Pourreza et al., 2024), Distillery (Maamari et al., 2024), and CHESS (Talaei et al., 2024). From the results, on SPIDER, our ROUTE + Qwen2.5-14B achieves a similar performance as CHESS based on Gimini or GPt-4/4o. This suggests that our ROUTE is an exceptional choice in both conventional and privatized Text2SQL scenarios. On SPIDER, our ROUTE falls behind CHESS + proprietary by 5 points and CHASS-SQL + Gemini 1.5 by 12 points. We think this is because the database in BIRD is much more complex and contains a large number of tables and columns in a

Table 9: The performance (EX) comparison with different hardness on the BIRD development set. The best results in the fine-tuning group are highlighted in **bold**.

| Methods | Simple | Moderate | Challenging | All |
|---|---|---|---|---|
| *Prompting-based methods* | | | | |
| C3-SQL + ChatGPT (Dong et al., 2023) | 58.9 | 38.5 | 31.9 | 50.2 |
| DAIL-SQL + GPT-4 (Gao et al., 2024a) | 62.5 | 43.2 | 37.5 | 54.3 |
| DAIL-SQL + GPT-4 + Self-consistency | 63.0 | 45.6 | 43.1 | 55.9 |
| SuperSQL + GPT-4 (Li et al., 2024a) | 66.9 | 46.5 | 43.8 | 58.5 |
| MAC-SQL + GPT-4 (Wang et al., 2023) | 65.7 | 52.7 | 40.3 | 59.4 |
| MCS-SQL + GPT-4 (Lee et al., 2024) | 70.4 | 53.1 | 51.4 | 63.4 |
| *Fine-tuning-based methods* | | | | |
| RESDSQL-3B (Li et al., 2023a) | 53.5 | 33.3 | 16.7 | 43.9 |
| CodeS-7B + SFT (Li et al., 2024b) | 64.6 | 46.9 | 40.3 | 57.0 |
| CodeS-15B + SFT (Li et al., 2024b) | 65.8 | 48.8 | **42.4** | 58.5 |
| **Ours:** Route + Llama3-8B | 64.3 | 49.3 | 36.8 | 57.3 |
| **Ours:** Route + Qwen2.5-7B | 63.8 | 45.4 | 39.6 | 55.9 |
| **Ours:** Route + Qwen2.5-14B | **67.7** | **53.1** | **42.4** | **60.9** |

Table 10: The performance (EX) of different open-source LLMs on SPIDER and BIRD development sets. ♡ and ◇ are used to mark the methods using the same base LLMs.

| Method | SPIDER | BIRD |
|---|---|---|
| ♡ MAC-SQL + SQL-Llama7B (Wang et al., 2023) | 76.3 | 43.9 |
| ♡ Sense-7B (Yang et al., 2024b) | 83.2 | 51.8 |
| ◇ DTS-SQL (Pourreza & Rafiei, 2024b) | 82.7 | 55.8 |
| Qwen2-7B + SFT | 78.0 | 45.9 |
| Qwen2-7B + SFT + MCP | 81.3 | 50.9 |
| Qwen2-7B + MSFT | 79.4 | 47.9 |
| Qwen2-7B + Route | 82.3 | 54.1 |
| CodeLlama-7B + SFT | 79.4 | 45.6 |
| CodeLlama-7B + SFT + MCP | 81.1 | 50.1 |
| CodeLlama-7B + MSFT | 79.7 | 46.4 |
| ♡ CodeLlama-7B + Route | 83.2 | 52.2 |
| Deepseek-Coder-7B + SFT | 81.2 | 51.2 |
| Deepseek-Coder-7B + SFT + MCP | 81.7 | 55.5 |
| Deepseek-Coder-7B + MSFT | 81.8 | 52.1 |
| ◇ Deepseek-Coder-7B + Route | 83.7 | 57.0 |
| Qwen2.5-Coder-7B + SFT | 82.8 | 51.2 |
| Qwen2.5-Coder-7B + SFT + MCP | 84.2 | 54.6 |
| Qwen2.5-Coder-7B+ MSFT | 82.0 | 50.7 |
| Qwen2.5-Coder-7B + Route | 84.3 | 57.8 |

single database leading to a very long context in input. It is widely acknowledged that small-sized LLMs (7B or 14B) are relatively limited in reasoning capabilities and handling long texts.

## A.4 PERFORMANCE ON SCHEMA LINKING

In this appendix, like (Pourreza & Rafiei, 2024b), we report the performance (Recall and Precision) of our schema linking module. The results shown in Table 12 demonstartes several important observations:

| Methods | SPIDER-Dev-EX | BIRD-Dev-EX | BIRD-Dev-VES |
|---|---|---|---|
| CHASE-SQL + Gemini 1.5 | 87.6 | 73.1 | 73.0 |
| Distillery + GPT-4o | - | 67.2 | 72.9 |
| CHESS + proprietary (GPT-4) | 87.2 | 65.0 | 65.4 |
| ROUTE + Qwen2.5-14B | 87.3 | 60.8 | 65.2 |

Table 11: The comparisons with recent methods on SPIDER and BIRD.

- After MSFT, the schema linking capability is significantly improved, especially in terms of precision scores.
- A higher Recall score generally leads to improved EX performance due to minor information loss.
- While simplifying the database schema is necessary, ensuring its completeness is more crucial for achieving enhanced performance.

| $SL_{\sigma_s}$ | $SL_{\sigma_t}$ | SPIDER | | | | | BIRD | | | | |
|---|---|---|---|---|---|---|---|---|---|---|---|
| | | EX | T-R | T-P | C-R | C-P | EX | T-R | T-P | C-R | C-P |
| ✓ | ✓ | 85.8 | **98.75** | 94.67 | **99.24** | 96.36 | 56.0 | **95.21** | 88.50 | **96.95** | 89.93 |
| ✓ | | 84.1 | 97.38 | 95.71 | 98.59 | 96.98 | 52.7 | 95.23 | 90.10 | 88.21 | 91.86 |
| | ✓ | 85.0 | 97.01 | 97.26 | 98.21 | 97.99 | 54.5 | 91.60 | 93.14 | 94.15 | 94.11 |
| | | 83.3 | 100.00 | 18.27 | 100.00 | 40.05 | 53.1 | 100.00 | 12.23 | 100.00 | 31.64 |
| ✓ | ✓ | 73.3 | **97.43** | 74.99 | **98.61** | 90.33 | 36.8 | **93.65** | 73.57 | **95.78** | 84.36 |
| ✓ | | 64.5 | 88.35 | 76.37 | 94.83 | 91.46 | 30.4 | 83.77 | 75.38 | 89.55 | 86.39 |
| | ✓ | 73.1 | 94.24 | 91.60 | 97.12 | 95.30 | 35.4 | 82.15 | 89.01 | 88.32 | 91.63 |
| | | 69.3 | 100.00 | 38.47 | 100.00 | 18.12 | 32.1 | 100.00 | 12.23 | 100.00 | 31.64 |

Table 12: The performance of schema linking. T-R/P means the Recall/Precision scores on table linking and C-R/P means the Recall/Precision scores on column linking. The best Recall scores are in **bold**. The first four rows are the results of the LLM after MSFT, and the last four rows are the results of the original LLM.

## A.5 THE RESULTS ON DR.SPIDER

In this appendix, we conduct experiments on Dr.Spider (Chang et al., 2023) to have a clearer and more comprehensive understanding of the advantages of our ROUTE. Dr.Spider includes 17 perturbation variants that can comprehensively measure the effectiveness and robustness. The specific experimental results are shown in Tables 13 and 14, including the study on MSFT and on each component. In the table, Avg.DB means the average results on DB perturbation test sets, Avg.NLQ means the average results on NLQ perturbation test sets, Avg.SQL means the average results on SQL perturbation test sets, and Avg.all means the average results on all test sets. From the results, the conclusions are consistent with those on SPIDER and BIRD and each component brought performance improvement, which shows that each task has an indispensable contribution to the performance. This further verifies the advantages and robustness of ROUTE.

## A.6 THE IMPACT OF SINGLE TASK SFT

In this appendix, we conduct additional experiments to explore the impact of single/triple-task SFT and MSFT on each task. For Text-to-SQL (TS), we report zero-shot EX results on the SPIDER and BIRD development sets. For Schema Linking (SL), we report the Recall/Precession scores of predicted related tables and columns. For Noise Correction (NC), we reports the EX scores of SQL queries refined with Noise Correction on the output SQL queries of Llama3-8B. For Continuation Writing (CW), we reports the EX scores of all SQL queries obtained by continuation writing on half of ground-truth SQL queries. The specific experimental results are shown in following table, where '−' means that LLMs cannot obtain the output in the expected format due to overfitting. The

| Methods | Avg.DB Pre~Post | Avg.NLQ Pre~Post | Avg.SQL Pre~Post | Avg.all Pre~Post |
|---|---|---|---|---|
| Llama3 | 70.1~54.3 | 70.6~56.8 | 69.1~65.5 | 69.9~58.8 |
| Llama3 + MCP | 75.6~59.1 | 77.0~61.7 | 74.8~72.6 | 75.8~64.4 |
| Llama3 + SFT | 83.4~66.0 | 83.0~72.8 | 79.9~77.6 | 82.1~72.2 |
| Llama3 + SFT + MCP | 85.4~68.0 | 85.3~75.1 | 84.3~81.7 | 85.0~74.9 |
| Llama3 + MSFT | 83.8~66.3 | 82.9~72.5 | 80.0~77.5 | 82.2~72.1 |
| **Llama3 + ROUTE** | **86.7~69.6** | **85.4~75.8** | **84.5~81.9** | **85.5~75.8** |

Table 13: The performance on Dr.Spider benchmark. The best results are highlighted in **bold**.

| No. | SL | NC | CW | Avg.DB Pre-Post | Avg.NLQ Pre-Post | Avg.SQL Pre-Post | Avg.all Pre-Post |
|---|---|---|---|---|---|---|---|
| #1 | ✓ | ✓ | ✓ | 86.7~69.6 | 85.4~75.8 | 84.5~81.9 | 85.5~75.8 |
| #2 | ✓ | | | 86.3~67.9 | 84.6~75.4 | 84.4~82.1 | 85.1~75.1 |
| #3 | | ✓ | | 84.4~67.7 | 83.7~73.7 | 80.3~78.7 | 82.8~73.3 |
| #4 | | | ✓ | 84.0~66.6 | 83.0~73.0 | 80.1~78.0 | 82.4~72.6 |
| #5 | | | | 83.8~66.6 | 82.9~72.5 | 80.0~77.5 | 82.2~72.1 |

Table 14: The ablation results (EX) on Dr.Spider.

results suggest that tasks not included in MSFT (*e.g.*, No.#2-9) demonstrate lower performance due to the overfitting of LLMs to other tasks, which is not conducive to multi-task collaboration. This highlights the importance of considering SFT across multiple tasks to prevent performance degradation of an LLM when handling additional tasks. On the contrary, although the performance of the full MSFT on certain tasks, such as SL and CW, is somewhat inferior to that of single or triple-task SFT, the full MSFT (No.#1) demonstrates significant performance improvements across each task and exhibits better stability, which reduces the risk of overfitting, thus improving the feasibility of multi-tasking collaboration.

| No. | Settings | TS EX EX | SPIDER-SL Table-R/P | SPIDER-SL Column-R/P | BIRD-SL Table-R/P | BIRD-SL Column-R/P | NC EX EX | CW EX EX |
|---|---|---|---|---|---|---|---|---|
| #1 | **MSFT** | **83.6 53.6** | **97.38/95.71** | **98.59/96.98** | **90.87/90.22** | **96.13/90.89** | **83.4 53.4** | **91.1 73.9** |
| #2 | MSFT w/o TS | 0.1 16.2 | 96.58/93.94 | 98.40/96.32 | 90.79/88.26 | 95.95/90.34 | 77.4 45.5 | 86.5 69.6 |
| #3 | MSFT w/o SL | 81.8 50.9 | – | – | – | – | 76.3 47.4 | 91.3 73.5 |
| #4 | MSFT w/o NC | 82.8 51.0 | 96.52/94.25 | 99.00/96.59 | 90.41/88.85 | 96.09/90.75 | – – | 91.7 73.4 |
| #5 | MSFT w/o CW | 81.2 50.3 | 96.51/93.97 | 98.65/96.39 | 90.59/88.12 | 96.05/90.64 | 79.4 49.0 | 81.2 56.7 |
| #6 | SFT with TS | 83.1 52.9 | – | – | – | – | – – | 85.6 69.2 |
| #7 | SFT with SL | – – | 95.55/92.69 | 98.91/95.29 | 87.84/85.11 | 94.93/89.51 | – – | – – |
| #8 | SFT with NC | 0.1 8.7 | – | – | – | – | 78.9 49.3 | 48.6 38.6 |
| #9 | SFT with CW | 68.1 39.0 | – | – | – | – | – – | 89.8 70.1 |
| #10 | Llama3 w/o SFT | 69.3 32.1 | 88.35/76.37 | 94.83/91.46 | 83.77/75.38 | 89.55/86.39 | 72.1 38.1 | 80.3 57.6 |

Table 15: The SFT impact of all tasks on each other. The results of MSFT are in **bold**.

## A.7 DETAILS OF NOISY CORRESPONDENCE FILTERING

In this appendix, we mainly clarify some details in noisy correspondence filtering. To obtain more diversity negative examples, we artificially introduce some errors to the ground-truth SQL queries. Based on some recent works exploring, we focus on five types of errors, schema linking errors, nesting errors, GROUP BY errors, JOIN errors, and symbol errors. More specifically,

- Schema linking error refers to the wrong table and column names in the SQL query. We randomly introduce such errors by making typos and synonym substitutions for table or column names.

- Nesting errors refer to the need to use nested or set operations but not using them (*e.g.*, `UNION`, `UNION ALL`, `INTERSECT` and `EXCEPT`). We destroy the SQL queries by randomly removing the sub-SQLs before or after these keywords.

- JOIN errors commonly occur in that the SQL queries using `JOIN` operation focus on the wrong table or column names. We introduce such errors by randomly replacing the table or column names.

- GROUP BY errors commonly occur in that the SQL queries using `GROUP BY` operation focus on the wrong column names. We introduce such errors by randomly replacing the column names after `GROUP BY`.

- Symbol errors are some minor errors such as incorrect keywords, missing commas, missing parentheses, and confusing function names (such as `COUNT`, `MAX`, `MIN`, *etc.*).

To obtain artificially constructed negative examples, we select a certain type of error according to a certain probability and introduce it into the ground-truth SQL queries. If the SQL query does not meet the type of error introduced, such as no GROUP BY operation, we will randomly select other types of errors to continue to construct negative examples.

### A.8 INNOVATION AND DIFFERENCE DISCUSSION

In this appendix, we provide more discussion to further elaborate on the innovations of our ROUTE and highlight how it differs from existing methods. The valuable insights and significant contributions of our work can be summarized as follows:

- ROUTE is among the pioneering frameworks under the context of LLMs that explores multi-task tuning and collaborative prompting to improve Text2SQL performance.

- We have exhaustively introduced and defined three important tasks in SQL generation, demonstarting that multi-task tuning and collaborative prompting in Schema Linking, Noise Correction and Continuation Writing significantly improve SQL generation accuracy. The additionally introduced SQL-related tasks are well integrated during both the training and inference phases.

- We have achieved state-of-the-art performance in 7B/14B-sized LLMs on both the widely-recognized SPIDER and BIRD benchmarks, with verified generalization and transferability across various cross-domains benchmarks and LLMs.

We highlight the key similarities and differences between our ROUTE and other related works as follows:

1. **Multi-task Supervised Fine-tuning (MSFT)**: The method most comparable to our ROUTE approach is MAC-SQL (Wang et al., 2023), which introduces multiple task agents and demonstrates the effectiveness of fine-tuning through the use of multi-agent instructions on CodeLlama-7B (Roziere et al., 2023).

   (a) First, the defined tasks in MSFT for ROUTE differ from those in MAC-SQL. We have introduced a new continuation writing (CW) task to further refine the challenging SQL queries. As demonstrated in Table 15, CW holds significant potential for SQL generation. On SPIDER development set, exploring CW task is able to achieve an impressive EX score of 91.1.

   (b) Second, in MAC-SQL, generating instruction data for SFT involves decomposing complex questions into multiple sub-questions and constructing corresponding answers. In contrast, our approach, beyond noise correction, allows for the synthesis of SFT data for various tasks using programming functions. This makes our method more practical for large-scale multi-task data synthesis for MSFT.

   (c) Third, in terms of performance, our ROUTE is significantly outperforms MAC-SQL based on the open-source LLM of CodeLlama-7B. The detailed results are presented in Table 10.

2. **SQL-Data Synthesis**: Our ROUTE involves the synthesis of SQL-related instruction-following data, which shares similarities with the recent work SENSE (Yang et al., 2024b).

   (a) First, compared to SENSE, the data synthesis pipeline of ROUTE encompasses not only Text2SQL but also multiply other SQL-related tasks. Our approach focuses on utilizing

existing data to synthesize multi-task SFT data, thereby enhancing the capabilities of open-source LLMs to handle various SQL-related tasks. In contrast, SENSE mainly focused on SQL generation task, leveraging strong LLMs to increase the diversity of SQL generation training set and synthesize preference data.

(b) Besides, our ROUTE achieves comparable performance to SENSE on the SPIDER development set and better performance on the BIRD development set, as shown in Table 10.

3. **Multi-tasking Collaboration**: To exploit the potential of multi-task capabilities, we propose a multi-task collaborative prompting strategy (MCP) to improve the final SQL generation. The most similar works are DIN-SQL (Pourreza & Rafiei, 2024a) and MAC-SQL (Wang et al., 2023), which both aim to reduce the complexity of the Text2SQL and improve the final performance via self-correction.

(a) First, compared to them, our MCP efficiently integrates multiple tasks using concise prompts across all tasks, which makes it more effective in small-sized LLMs that struggle with comprehending complex instructions. As shown in the results of Table 1, the effectiveness of MAC-SQL and DIN-SQL is constrained by the limited capacity of small-sized LLMs to comprehend complex instructions, while our MCP can achieve better and impressive performance.

(b) Besides, while all three methods employ a self-correction strategy to enhance the quality of generated SQL queries, our MCP introduces a novel continuation writing task specifically designed to refine challenging SQL queries and improve the performance significantly.

Considering the comprehensive nature of our work, which encompasses data synthesis, supervised fine-tuning, and multi-task collaborative prompting, it is inevitable that there are some similarities with existing work. Nevertheless, our method has offered numerous insights into the Text2SQL and achieved promising results, which we believe are significant contributions to the Text2SQL community.

## A.9 Algorithm of MCP

In this appendix, to make our MCP clearer, we describe the pipeline in detail in Algorithm 1.

---

**Algorithm 1** The algorithm of MCP

---

**Input:** The database $d$, user question $q$, and LLM $\mathcal{M}$;
    // Conduct schema linking.
1: Obtain simplified database $\tilde{d}$ via Equation (2);
    // SQL generation.
2: Generate intermediate SQL query $s^*$ via $\mathcal{M}(\sigma_t(\tilde{d}, q))$;
    // Conduct noise correction.
3: Check the SQL query $s^*$ via $\mathcal{M}(\sigma_n(d, q, s^*, e))$.
4: Obtain the the correct SQL $\tilde{s}$ if $\mathcal{M}$ shows that $s^*$ is inaccurate.
5: **if** $\mathcal{M}$ shows $s^*$ is inaccurate and $\mathrm{SQLer}(d, \tilde{s})$ is True **then**
6:     $s^* = \tilde{s}$.
7: **end if**
    // Refine wrong or hard SQL queries by continuation writing.
8: **if** $\mathrm{SQLer}(d, s^*)$ is False or $h(s^*, d) > 2$ **then**
9:     Construct the truncated SQL query $\hat{s}$ based on $s^*$;
10:     Continue writing: $\bar{s} = \mathcal{M}(\sigma_c(d, q, \hat{s}))$;
11:     **if** $\mathrm{SQLer}(d, \bar{s})$ is True **then**
12:         $s^* = \bar{s}$;
13:     **end if**
14: **end if**
**Output:** The final SQL query $s^*$.

---

A.10    STATISTICS OF MSFT DATA

Our MSFT data includes the main SFT data for Text2SQL and the SFT data for three other tasks. The former consists of the SPIDER and BIRD training sets after noisy correspondence filtering (898 pairs removed), with a total of 15,530 pairs. The SFT data for other tasks are randomly selected from the filtered dataset and 10,000 data pairs are constructed. That is to say, our MSFT data has a total of 45,530 data pairs.

A.11    PROMPT TEMPLATES

In this appendix, we provide the prompts of all tasks used in our pipeline as shown in Figures 5 to 8. Among these prompts, only the BIRD dataset provides question hints. Meanwhile, other information, such as few-shot examples and execution exceptions, only provides for non-SFT models to guide the output. The prompt used for noisy correspondence filtering is presented in Figure 9, which is similar to the NC's instructions (Figure 6) but does not include the terms of execution exception to avoid LLMs focusing only on the wrong pattern instead of the discriminative pattern.

---

Given the following database schema and question, your task is to write a valid SQL query whose execution results can accurately answer the question.

/* Database schema */
CREATE TABLE customers (CustomerID NUMBER, Segment TEXT, Currency TEXT, PRIMARY KEY(CustomerID, Segment, Currency)); CREATE TABLE transactions_1k (CustomerID NUMBER, Amount NUMBER, Price NUMBER, PRIMARY KEY(CustomerID, Amount, Price));

/* Sample rows of each table */
customers: [(3, 'SME', 'EUR'), (5, 'LAM', 'EUR'), (6, 'SME', 'EUR')]
transactions_1k: [(31543, 28, 672.64), (46707, 18, 430.72), (46707, 1, 121.99)]

/* Question */
Who is the top spending customer and how much is the average price per single item purchased by this customer? What currency was being used?

/* Question hint */
verage price per single item = price / amount

Answer the question by a SQL query only with no explanation:

---

Figure 5: The prompt of Text-to-SQL.

Your task is to determine whether the execution results of a SQL query can answer the given question according to the following database schema. If the execution results cannot correctly answer the question, please give me the correct SQL query.

/* Examples */
...few-shot examples omitted...

/* Database schema */
CREATE TABLE customers (CustomerID NUMBER, Segment TEXT, Currency TEXT, PRIMARY KEY(CustomerID, Segment, Currency)); CREATE TABLE transactions_1k (CustomerID NUMBER, Amount NUMBER, Price NUMBER, PRIMARY KEY(CustomerID, Amount, Price));

/* Sample rows of each table */
customers: [(3, 'SME', 'EUR'), (5, 'LAM', 'EUR'), (6, 'SME', 'EUR')]
transactions_1k: [(31543, 28, 672.64), (46707, 18, 430.72), (46707, 1, 121.99)]

/* Question */
Who is the top spending customer and how much is the average price per single item purchased by this customer? What currency was being used?

/* Question hint */
verage price per single item = price / amount

/* SQL query */
SELECT T2.CustomerID, SUM(T2.Price / T2.Amount), T1.Currency FROM customers AS T1 INNER JOIN transactions_1k AS T2 ON T1.CustomerID = T2.CustomerID WHERE T2.CustomerID = (SELECT CustomerID FROM yearmonth ORDER BY Consumption DESC LIMIT 1) GROUP BY T2.CustomerID, T1.Currency

/* Execution exception */
...information omitted...

Output:

Figure 6: The prompt template of noise correction.

Given the following database schema and question, your task is to extract the tables and columns relevant to solving the question.

/* Examples */
...few-shot examples omitted...

/* Database schema */
CREATE TABLE customers (CustomerID NUMBER, Segment TEXT, Currency TEXT, PRIMARY KEY(CustomerID, Segment, Currency)); CREATE TABLE transactions_1k (CustomerID NUMBER, Amount NUMBER, Price NUMBER, PRIMARY KEY(CustomerID, Amount, Price));

/* Sample rows of each table */
customers: [(3, 'SME', 'EUR'), (5, 'LAM', 'EUR'), (6, 'SME', 'EUR')]
transactions_1k: [(31543, 28, 672.64), (46707, 18, 430.72), (46707, 1, 121.99)]

/* Question */
Who is the top spending customer and how much is the average price per single item purchased by this customer? What currency was being used?

/* Question hint */
verage price per single item = price / amount

Output:

Figure 7: The prompt template of schema linking.

Given the following database schema and question, your task is to write an incomplete SQL query into a complete SQL query whose execution results can correctly answer the question.

/* Examples */
...few-shot examples omitted...

/* Database schema */
CREATE TABLE customers (CustomerID NUMBER, Segment TEXT, Currency TEXT, PRIMARY KEY(CustomerID, Segment, Currency)); CREATE TABLE transactions_1k (CustomerID NUMBER, Amount NUMBER, Price NUMBER, PRIMARY KEY(CustomerID, Amount, Price));

/* Sample rows of each table */
customers: [(3, 'SME', 'EUR'), (5, 'LAM', 'EUR'), (6, 'SME', 'EUR')]
transactions_1k: [(31543, 28, 672.64), (46707, 18, 430.72), (46707, 1, 121.99)]

/* Question */
Who is the top spending customer and how much is the average price per single item purchased by this customer? What currency was being used?

/* Question hint */
verage price per single item = price / amount

/* The incomplete SQL query */
```sql
SELECT T2.CustomerID, SUM(T2.Price / T2.Amount), T1.Currency FROM customers AS T1 INNER JOIN transactions_1k AS T2 ON
```

Output:

Figure 8: The prompt template of continuation writing.

Your task is to determine whether the execution results of a SQL query can answer the given question according to the following database schema. If the execution results cannot correctly answer the question, please give me the correct SQL query.

/* Database schema */
CREATE TABLE customers (CustomerID NUMBER, Segment TEXT, Currency TEXT, PRIMARY KEY(CustomerID, Segment, Currency)); CREATE TABLE transactions_1k (CustomerID NUMBER, Amount NUMBER, Price NUMBER, PRIMARY KEY(CustomerID, Amount, Price));

/* Sample rows of each table */
customers: [(3, 'SME', 'EUR'), (5, 'LAM', 'EUR'), (6, 'SME', 'EUR')]
transactions_1k: [(31543, 28, 672.64), (46707, 18, 430.72), (46707, 1, 121.99)]

/* Question */
Who is the top spending customer and how much is the average price per single item purchased by this customer? What currency was being used?

/* Question hint */
verage price per single item = price / amount

/* SQL query */
SELECT T2.CustomerID, SUM(T2.Price / T2.Amount), T1.Currency FROM customers AS T1 INNER JOIN transactions_1k AS T2 ON T1.CustomerID = T2.CustomerID WHERE T2.CustomerID = (SELECT CustomerID FROM yearmonth ORDER BY Consumption DESC LIMIT 1) GROUP BY T2.CustomerID, T1.Currency

Output:

Figure 9: The prompt of STF data for noisy correspondence filtering.

## A.12 QUALITATIVE EXAMPLES OF NOISY PAIRS

In Listings 1 and 2, we provide more identified noisy pairs in SPIDER and BIRD training sets. To show the error visually, we mainly provide noisy examples with obvious semantic inconsistencies.

```
Q1: /*How many followers does each user have?*/
A1: SELECT count(*) FROM follows;
R1: SELECT count(*), T1.name FROM user_profiles AS T1 JOIN follows AS T2
    ON T1.uid = T2.f1 GROUP BY T1.uid;

Q2: /*Find the number of followers for each user.*/
A2: SELECT count(*) FROM follows GROUP BY f1
R2: SELECT count(*), f1 FROM follows GROUP BY f1;

Q3: /*What is the party that has the largest number of representatives?*/
A3: SELECT Party FROM representative GROUP BY Party ORDER BY COUNT(*)
    DESC LIMIT 1;
R3: SELECT Party,  COUNT(*) FROM representative GROUP BY Party ORDER BY
    COUNT(*) DESC LIMIT 1;

Q4: /*Find the number of kids staying in the rooms reserved by a person
    called ROY SWEAZ.*/
A4: SELECT kids FROM Reservations WHERE FirstName = "ROY" AND LastName =
    "SWEAZY";
R4: SELECT sum(T2.Kids) FROM Rooms AS T1 JOIN Reservations AS T2 ON T1.
    RoomId = T2.Room WHERE T2.FirstName = "ROY" AND T2.LastName = "SWEAZ"
    ;

Q5: /*Which manufacturer has the most number of shops? List its name and
    year of opening.*/
A5: SELECT open_year, name FROM manufacturer ORDER BY num_of_shops DESC
    LIMIT 1;
R5: SELECT name, open_year FROM manufacturer ORDER BY num_of_shops DESC
    LIMIT 1;
```

Listing 1: Identified noisy pairs in SPIDER training set.

```
Q1: /*What is the number of inhabitants and income of geographic
    identifier 239?*/
A1: SELECT INHABITANTS_K FROM Demog WHERE GEOID = 239;
R1: SELECT INHABITANTS_K, INCOME_K FROM Demog WHERE GEOID = 239;

Q2: /*List the geographic id of places where the income is above average.
    */
A2: SELECT AVG(INCOME_K) FROM Demog;
R2: SELECT GEOID FROM Demog WHERE INCOME_K > (SELECT AVG(INCOME_K) FROM
    Demog);

Q3: /*Average length of the rivers flowing into the Donau River.*/
A3: SELECT * FROM river WHERE Name = 'Donau'
R3: SELECT avg(Length) FROM river WHERE Name = 'Donau';

Q4: /*What are the corresponding classes for the "very large bike"
    attribute? */
A4: SELECT ATT_CLASS_ID FROM ATT_CLASSES WHERE ATT_CLASS = 'very_large';
R4: SELECT ATT_CLASS_ID FROM ATT_CLASSES WHERE ATT_CLASS = 'very_large_
    bike';

Q5: /*Which Shakespeare story with character ID 324 has description of '
    this friend of Caesar'?*/
A5: SELECT T1.Title FROM works AS T1 INNER JOIN chapters AS T2 ON T1.id =
     T2.work_id INNER JOIN paragraphs AS T3 ON T2.id = T3.chapter_id
    INNER JOIN characters AS T4 ON T3.character_id = T4.id WHERE T2.id =
    '324' AND T2.Description = 'friend_to_Caesar';
```

```
R5: SELECT T1.Title FROM works AS T1 INNER JOIN chapters AS T2 ON T1.id =
    T2.work_id INNER JOIN paragraphs AS T3 ON T2.id = T3.chapter_id
    INNER JOIN characters AS T4 ON T3.character_id = T4.id WHERE T3.
    character_id = 324 AND T4.Description = 'this friend of Caesar';
```

Listing 2: Identified noisy pairs in BIRD training set.

### A.13 ETHICS AND REPRODUCIBILITY STATEMENT

This paper aims to explore the possibility of improving Text2SQL performance by the multitask collaboration based on large language models. For this purpose, we use some open-source LLMs as our investigated models to propose solutions, which may pose potential commercial risks. We pledge that we will only conduct academic research on these LLMs to verify the effectiveness of our proposed solution and will not use them for other purposes.

In addition, to ensure the reproducibility of our research, we have made several efforts to ensure that our solution is convincing. First, we provide details in Section 3.2 to clarify the data construction pipeline used for MSFT. In addition, the used prompt templates are provided in Appendix A.11 to ensure reproducibility. During the training stage, we utilized a unified fine-tuning framework, *i.e.*, Llama-Factory, and detailed the settings of core parameters in Section 4.1. During the inference stage, we recommend setting a very low temperature of $0.01$ to ensure the reproducibility of LLMs. Finally, we have released the code and synthetic data at here for reproducibility, thus further advancing the Text2SQL community.

