# OpenReview forum: "ROUTE: Robust Multitask Tuning and Collaboration for Text-to-SQL"
_ICLR.cc/2025/Conference — ICLR 2025 Poster_

### Official Review · Reviewer_Ej7x · 2024-10-30

**Soundness:** 3
**Presentation:** 3
**Contribution:** 2
**Rating:** 6
**Confidence:** 4

**Summary:**

The paper proposes ROUTE, a method that involves (1) multi-task training and (2) multi-stage prompting to improve LLMs’ performance on text-to-SQL parsing. Compared to an extensive list of baselines, the proposed ROUTE method demonstrates better parsing accuracy on two established datasets, Spider and BIRD, and three other perturbed variants of Spider.

**Strengths:**

1. The performance of ROUTE is strong. On two well-established text-to-SQL datasets, Spider and Bird, ROUTE effectively improves the performance of two latest open-weight LLMs, Llama 3.1 and Qwen 2.5, and achieves comparable performance to GPT-4. The performance improvement also holds on three perturbed Spider variants, indicating the robustness of ROUTE.
2. The experiments to evaluate ROUTE are comprehensive. The authors gathered an extensive list of baselines and compared them with ROUTE (or the multi-stage prompting step in ROUTE). Additional experiments and ablation studies of the method further supports some design choices of ROUTE and demonstrates its stable performance improvement across different models and datasets.
3. The paper is easy-to-follow, and the writing is mostly clear.

**Weaknesses:**

1. It is not very clear what kind of novel contribution this paper is making. The tasks themselves for multi-task training and multi-stage prompting have all been studied in related work, and some of them are rebranded under new terms. To name some examples, “noise correction” is essentially training and prompting LLMs to self-debug [1][2], and the “continuation writing” is simply a subset of text-to-SQL generation by the autoregressive nature of LLMs and has been one of the prompting paradigm for text-to-SQL parsing with LLMs [3][4]. At the framework level, there are also existing papers compiling different tasks to improve LMs’ text-to-SQL performance via multi-task training [5]. Thus, it is opaque how the proposed method combines these existing ideas in a novel way.

2. The noisy correspondence filtering step to pre-process the training data is not fully elaborated, and the contribution of this step to ROUTE is minimal according to the ablation study (#1 vs #3 and #6 vs #8 in Table 3). Training details and quality of the noise filtering model is not discussed, e.g. through an intrinsic evaluation of how accurately it can discriminate noisy examples. The difference between ROUTE’s data synthesis procedure and that of SENSE is not clear. The method to “artificially and randomly introduce errors” (lines 225-230) is also not documented. Overall, this part of the method is not clearly explained, and its contribution in ROUTE is not obvious.

[1] https://arxiv.org/abs/2304.05128

[2] https://arxiv.org/abs/2312.11242

[3] https://arxiv.org/abs/2204.00498

[4] https://arxiv.org/abs/2303.13547

[5] https://arxiv.org/abs/2212.09278

**Questions:**

1. What is the “pseudo-SQL” used to perform schema linking? How is it implemented? This term only appears twice in the paper without any further elaboration.

2. The use of “hallucination” may not be appropriate here in the context of text-to-SQL parsing. Are the authors simply trying to say incorrect column matching and entity linking?

3. The manuscript would benefit from another round of proof-read to correct typos and standardize term usage, including those mentioned above and some other examples as follows:
- “in-contextual learning” -> “in-context learning” (line 39)
- “promoting-based methods” -> “prompting-based methods” (lines 200, 373)
- “shema linking” -> “schema linking” (line 228)
 - “SQLer$(d_i, \tilde{s}^*)$” -> “SQLer$(d_i, s^*)$” (line 283)

---

> ### Author Response · Authors · 2024-11-23
> **Official reply to Reviewer Ej7x [1/2]**
>
> We thank Reviewer Ej7x for the positive recognition of our ROUTE performance and the insightful comments that contribute to improving our paper. Below, we address your concerns one by one.
>
> >W1: It is not very clear what kind of novel contribution this paper is making. The tasks themselves for multi-task training and multi-stage prompting have all been studied in related work, and some of them are rebranded under new terms. To name some examples, “noise correction” is essentially training and prompting LLMs to self-debug [1][2], and the “continuation writing” is simply a subset of text-to-SQL generation by the autoregressive nature of LLMs and has been one of the prompting paradigm for text-to-SQL parsing with LLMs [3][4]. At the framework level, there are also existing papers compiling different tasks to improve LMs’ text-to-SQL performance via multi-task training [5]. Thus, it is opaque how the proposed method combines these existing ideas in a novel way.
>
> **Ans:** Thank you for your detailed review. We acknowledge that the proposed ROUTE shares similarities with several recent works [1,2,3,4,5]. However, it offers valuable insights and makes significant contributions to the field of Text2SQL, which can be summarized as follows:
>
> - ROUTE is among the pioneering frameworks under the context of LLMs that explores multi-task tuning and collaborative prompting to improve Text2SQL performance.
> - We have exhaustively introduced and defined three important tasks in SQL generation, demonstarting that multi-task tuning and collaborative prompting in Schema Linking, Noise Correction and Continuation Writing significantly improve SQL generation accuracy. The additionally introduced SQL-related tasks are well integrated during both the training and inference phases.
> - We have achieved state-of-the-art performance in 7B/14B-sized LLMs on both the widely-recognized SPIDER and BIRD benchmarks, with verified generalization and transferability across various cross-domains benchmarks and LLMs.
>
> We sumarize the differences and relationships between ROUTE and your mentioned works are as follows:\
> [1] proposed an effective Self-Debugging Strategy to teach existing LLMs to debug the predicted program via few-shot demonstrations, including SQL, Python, C++, etc. However, our method is self-debugging for SQL generation, and it is under the context of supervised fine-tuning. (**lines 101~102**)\
> [2] proposed a multi-agent collaboration framework and used the closed-source LLMs GPT-4 as a basis to enhance text-to-SQL parsing. However, we found that its migration to some small-sized models (Table 1) has limited effectiveness. We proposed MCP to achieve promising performance by combining multi-task collaboration with some simple instructions. (**lines 104~105**)\
> [3] performed an empirical evaluation based on Codex. However, it does not explicitly enhance the relevant capabilities by SFT, but only uses its autoregressive characteristics to achieve performance improvement with few-shot examples. Instead, we explored the solution that explicitly enhance multi-task capabilities to improve the accuracy of SQL generation. (**line 93**)\
> [4] is a pioneering work to evaluate the performance of ChatGPT on Text2SQL, showing that it has great potential for SQL generation. However, our paper focuses on open-source LLMs under the context of supervised fine-tuning. (**line 98**)\
> [5] proposed multiple subtasks and combined generative pretrained language models to improve Text2SQL. However, its focused subtasks do not completely overlap with ours. Moreover, the effectiveness of this paradigm in the context of LLMs is unknown. (**line 91**)
>
> >W2-I The noisy correspondence filtering step to pre-process the training data is not fully elaborated, and the contribution of this step to ROUTE is minimal according to the ablation study (#1 vs #3 and #6 vs #8 in Table 3).
>
> **Ans:** Thank you for your valuable comment. For SFT, the impact of noisy correspondence filtering (#6 vs #8) is not obvious, which indicates that noisy data accounts for a relatively low proportion in the Spider and Bird datasets. After introducing multi-task data synthesis, we observed that the noisy correspondence filtering step significantly boosts performance (#1 vs #3) on SPIDER. This indicates that the noise is substantial, and the accumulation of noise from multi-task data synthesis can adversely impact the model's comprehension of basic (simple) modes.

---

> ### Author Response · Authors · 2024-11-23
> **Official reply to Reviewer Ej7x [2/2]**
>
> >W2-II  **(1)** Training details and quality of the noise filtering model is not discussed, e.g. through an intrinsic evaluation of how accurately it can discriminate noisy examples.  **(2)** The difference between ROUTE’s data synthesis procedure and that of SENSE is not clear. **(3)** The method to “artificially and randomly introduce errors” (lines 225-230) is also not documented. **(4)** Overall, this part of the method is not clearly explained, and its contribution in ROUTE is not obvious.
>
> **Ans:** Thank you for your valuable comments, we will reply to this weakness point-by-point.\
> **(1)** According to the suggestion, we have added an experiment to assess the accuracy of noise sample identification. We construct an evaluation set from SPIDER and BIRD development sets to evaluate their ability to identify positive examples and negative examples, where the negative examples come from challenging artificially synthesized SQL queries and the incorrect SQL queries obtained by SQL generation with Llama3-8B.
>
> As shown in following table, we can observe that the LLM’s ability to identify negative (noisy) and positive examples has been significantly improved after SFT. Further, for more challenging dataset like Bird, there still remains room for improvement in distinguishing noisy samples.
>
> ||w/o SFT|w/o SFT|w/o SFT|with SFT|with SFT|with SFT |
> |-|-|-|-|:-:|:-:|:-:|
> ||Spider|Bird|All|Spider|Bird|All|
> | Positive (Ground-truth, 100)|0.51|0.23|0.37|0.96|0.83|0.90|
> | Negative (Artificial, 100)|0.64|0.50|0.57|0.68|0.77|0.73|
> | Negative (Llama3, 100)|0.63|0.83|0.73|0.96|0.94|0.95|
>
> **(2)**  Compared to SENSE, the data synthesis pipeline of ROUTE encompasses not only Text2SQL but also multiple other SQL-related tasks. Our approach focuses on utilizing existing data to synthesize multi-task SFT data, thereby enhancing the capabilities of open-source language models to handle various SQL-related tasks. In contrast, SENSE mainly focused on SQL generation task, leveraging powerful LLMs to increase the diversity of SQL generation training set and also synthesize preference data. We have clarified the relationship and difference between ROUTE and SENSE in our revised submission (**Appendix A.8**).
>
> **(3)** Thank you for your reminder, we have clarified it in **Appendix A.7** of our revised manuscript.
>
> **(4)** Please see our Answer to **W1**.
>
> >Q1: What is the “pseudo-SQL” used to perform schema linking? How is it implemented? This term only appears twice in the paper without any further elaboration.
>
> **Ans:** Thanks for your question. The pseudo SQL refers to the generated intermediate SQL using the defined template and the complete schema, i.e., $\mathcal{M}(\sigma_t(d_i, q_i),d_i)$, which we have clarified in the revised submission (please see **lines 277~284** for the details)
>
> >Q2: The use of "hallucination" may not be appropriate here in the context of text-to-SQL parsing. Are the authors simply trying to say incorrect column matching and entity linking?
>
> **Ans:** Thanks for your feedback. We follow several recent works to utilize hallucinations in SQL generarion task [6,7]. Hallucinations in LLMs refer to cases where LLM generate plausible but factually incorrect or nonsensical information[8]. Hallucination in Text2SQL task indicates incorrect SQL generations. Schema hallucinations and logic hallucinations are widely observed in LLM-based SQL generation [7].
>
> Considering that the definition of hallucinations might be unclear in the context of Text2SQL, we have revised the usage of SQL hallucinations in the revised submission. Please see our revised submission for details.
>
> >Q3: The manuscript would benefit from another round of proof-read to correct typos and standardize term usage.
>
> **Ans:** Thanks for your suggestion. The typos and the standardize term usages have been double checked. We will re-polish our paper carefully.
>
> ### Reference
> [1] Teaching Large Language Models to Self-Debug, ICLR, 2024.\
> [2] MAC-SQL: A Multi-Agent Collaborative Framework for Text-to-SQL, arxiv preprint, 2024.\
> [3] Evaluating the Text-to-SQL Capabilities of Large Language Models, arxiv preprint, 2022.\
> [4] A comprehensive evaluation of ChatGPT's zero-shot Text-to-SQL capability, arxiv preprint, 2023.\
> [5] MIGA: A Unified Multi-task Generation Framework for Conversational Text-to-SQL, AAAI, 2023.\
> [6] Before Generation, Align it! A Novel and Effective Strategy for Mitigating Hallucinations in Text-to-SQL Generation. ACL 2024.\
> [7] PURPLE: Making a Large Language Model a Better SQL Writer. ICDE 2024.\
> [8] A Survey on Hallucination in Large Language Models: Principles, Taxonomy, Challenges, and Open Questions. ACM TOIS.

---

> > ### Comment · Reviewer_Ej7x · 2024-11-23
> > **Thanks for the Response**
> >
> > Thanks to the authors for their detailed response and efforts updating the paper. While I am not fully convinced, this paper may distinguish enough from related work as a novel combination of existing methods. Given the strong empirical performance and newly added experiments, I have increased my score accordingly.

---

> ### Author Response · Authors · 2024-11-24
>
> Dear Reviewer Ej7x,
>
> We greatly appreciate your timely feedback and positive support.  We will further refine our manuscript to improve its quality. Thanks again for your feedback and taking the time to consider our response.
>
> Best, Authors

---

> ### Author Response · Authors · 2024-11-27
>
> Dear Reviewer Ej7x,
>
> Thank you for raising the score! We have provided further clarification and discussion on the innovations, contributions, and differences between our ROUTE and existing methods in **Appendix A.8** of our revised submission.
>
> Best,
> Authors

---

### Official Review · Reviewer_TcaB · 2024-11-01

**Soundness:** 2
**Presentation:** 2
**Contribution:** 2
**Rating:** 5
**Confidence:** 5

**Summary:**

The paper introduces ROUTE, a method for enhancing text-to-SQL capabilities in open-source language models. The approach addresses limitations in current methods that rely heavily on closed-source large language models (LLMs), such as GPT-4, for text-to-SQL tasks. ROUTE leverages Multitask Supervised Fine-Tuning (MSFT) and Multitask Collaboration Prompting (MCP) to improve SQL generation performance by incorporating tasks like schema linking, noise correction, and continuation writing. These tasks enable a collaborative prompting approach that reduces hallucinations in SQL generation. Extensive experimentation on multiple benchmarks with open-source LLMs shows that ROUTE significantly improves SQL generation accuracy and outperforms recent methods using fine-tuning and prompting approaches.

**Strengths:**

1. The paper incorporates multiple tasks to enhance text-to-SQL capabilities, making the LLM more versatile and capable of handling complex SQL generation scenarios.
2. The paper evaluates ROUTE on several well-known benchmarks and compares its performance with other prompting and fine-tuning methods, demonstrating its effectiveness in real-world applications.

**Weaknesses:**

1. The authors mention that "Most training-based methods only incorporate the ⟨Question, SQL⟩ pairs for SFT, resulting in degraded performance in other tasks, such as schema linking." However, our approach usually incorporates a ⟨Question, Schema, SQL⟩ tuple for SFT. Additionally, a reduction in schema linking performance cannot be seen as a limitation of existing methods. If a specific task is not included in training, optimal results for that task are not expected. Therefore, this should not be considered a limitation; instead, one could state that training with schema linking can achieve better outcomes.
2. The authors state that "Training LLMs on a single SFT task poses a significant risk of overfitting, which may diminish the model's capability to understand instructions." However, overfitting is not further addressed in the subsequent sections. Could the authors clarify what overfitting entails in the context of SQL tasks, and explain how multi-task training specifically mitigates this risk? Additionally, training on more data and achieving good results may also suggest a potential overfitting scenario.
3. The authors mention that "This strategy leverages collaboration across several SQL-related tasks to reduce hallucinations during SQL generation." However, the term "SQL hallucinations" is not defined, nor is there any discussion in the experimental section explaining how hallucinations are reduced. This claimed advantage, therefore, remains unclear.
4. If Schema Linking, Noise Correction, and Continuation Writing are considered important, could the authors provide the relative improvement metrics for these tasks?
5. There are inconsistencies in writing style, such as using both "text-to-SQL" and "Text-to-SQL" interchangeably. Ensuring uniform terminology would improve the clarity and professionalism of the writing.

**Questions:**

1. The noise correction process assumes access to well-curated data and high-quality schema information, which might not be available for all databases or domains. Without rigorous data preparation, the model may struggle with hallucinations, as noise correction and schema linking effectiveness are diminished when data quality is compromised.
2.  In low-resource settings where high-quality SQL annotations or database schema information might be scarce, could ROUTE be enhanced by incorporating weak supervision, unsupervised learning, or semi-supervised data to fill gaps?
3. Given that database schemas often change over time in production, can ROUTE adapt to new tables or columns without needing extensive retraining, or would these require ongoing fine-tuning?

---

> ### Author Response · Authors · 2024-11-23
> **Official reply to Reviewer TcaB [1/3]**
>
> We thank Reviewer TcaB for the insightful comments and constructive suggestions that contribute to improving our paper. Below, we address your concerns one by one.
>
> >W1-I The authors mention that "Most training-based methods only incorporate the ⟨Question, SQL⟩ pairs for SFT, resulting in degraded performance in other tasks, such as schema linking." However, our approach usually incorporates a ⟨Question, Schema, SQL⟩ tuple for SFT.
>
> **Ans:** Thank you for your kindly reminder. We apologize for any confusion caused by our misleading expression of ⟨Question, SQL⟩. What we aim to say here is the ⟨Question, Schema, SQL⟩ tuple, as illustrated by our method's notions (**Section 3.1**) and the prompt template (**Appendix A.11**). We have revised this sentence in the revised submission. Thank you again for your feedback.
>
> >W1-II: Additionally, a reduction in schema linking performance cannot be seen as a limitation of existing methods. If a specific task is not included in training, optimal results for that task are not expected. Therefore, this should not be considered a limitation; instead, one could state that training with schema linking can achieve better outcomes.
>
> **Ans:** Thank you for your thoughtful comments.
>
> Schema linking is a critical step in the Text2SQL task [1,2], which demonstrates significant performance gains in SQL generation. Training exclusively on SQL generation tasks with SFT can severely impair the model's other capabilities (**Appendix A.6**), such as schema linking.
>
> Therefore, to maintain proficiency across various tasks for effective multi-task collaboration, it is essential to consider multi-task supervised fine-tuning.
>
> Considering the fact that SFT training in a single SQL generation task can significantly reduce its understanding of other instructions, thus making it unable to perform tasks like schema linking, we believe this is a potential drawback of the traditional SFT-based methods.
>
> To avoid potential misunderstanding, We have revsied this sentence as:
> **However, most training-based methods only incorporate the SQL generation task in the SFT stage, resulting in a degraded performance in other tasks that are important for Text2SQL capability, such as schema linking**
>
> >W2: The authors state that "Training LLMs on a single SFT task poses a significant risk of overfitting, which may diminish the model's capability to understand instructions." However, overfitting is not further addressed in the subsequent sections. Could the authors clarify what overfitting entails in the context of SQL tasks, and explain how multi-task training specifically mitigates this risk? Additionally, training on more data and achieving good results may also suggest a potential overfitting scenario.
>
> **Ans:** Thanks for your feedback. It has been widely observed that SFT training in a single task significantly reduces the model's instruction following ability on other tasks [3]. Overfitting on SQL generation task means that a fine-tuned LLM will weaken its ability to understand other instructions and perform poorly on other important SQL-related tasks (such as schema linking and noise correction).
>
> Therefore, in this paper, we explore multitask tuning across various SQL-related tasks to preserve the other essential capabilities of LLMs【see our submission in lines 63～69 for detailed description】We report the performance of single-task SFT on each task in the following table, where '--' means that LLMs cannot obtain the output in the expected format due to overfitting. The experimental results show that, compared to multi-task training, the models fine-tuned solely on a single task perform poorly on other tasks, which is caused by overfitting to the single task.
>
> **Table 15 in Appendix A.6:** The SFT impact of all tasks on each other.
>
> || |SPIDER-TS|BIRD-TS|SPIDER-SL|SPIDER-SL|BIRD-SL|BIRD-SL|SPIDER-NC|BIRD-NC|SPIDER-CW|BIRD-CW |
> |-|:-|:-:|:-:|:-:|:-:|:-:|:-:|:-:|:-:|:-:|:-:|
> |No.| Settings|Dev-EX|Dev-EX|Table-R/P|Column-R/P|Table-R/P|Column-R/P|Dev-EX|Dev-EX|Dev-EX|Dev-EX |
> | #1|Llama3 with MSFT|83.6 |53.6 |97.38/95.71|98.59/96.98|90.87/90.22|96.13/90.89|83.4 |53.4 |91.1 |73.9|
> | #2|SFT with TS|83.1 |52.9 |--|--|--|--|--|--|85.6 |69.2|
> | #3|SFT with SL|--|--|95.55/92.69|98.91/95.29|87.84/85.11|94.93/89.51 |--|--|--|-- |
> | #4|SFT with NC|0.1 |8.7 |--|--|--|--|78.9 |49.3 |48.6 |38.6|
> | #5|SFT with CW|68.1 |39.0 |--|--|--|--|--|--|89.8 |70.1|
> | #6|Llama3 w/o SFT|69.3 |32.1 |88.35/76.37|94.83/91.46|83.77/75.38|89.55/86.39|72.1 |38.1 |80.3 |57.6|
>
> According to your feedback, we have revised this sentence to minimize any confusion caused by the term "overfitting", as follows:\
> **Training LLMs on a single SQL generation task poses a substantial risk of diminishing performance in understanding instructions, potentially reducing the model's effectiveness in other important SQL-related tasks beyond SQL generation.**

---

> ### Author Response · Authors · 2024-11-23
> **Official reply to Reviewer TcaB [2/3]**
>
> >W3: The authors mention that "This strategy leverages collaboration across several SQL-related tasks to reduce hallucinations during SQL generation." However, the term "SQL hallucinations" is not defined, nor is there any discussion in the experimental section explaining how hallucinations are reduced. This claimed advantage, therefore, remains unclear.
>
> **Ans:** We follow several recent works to utilize hallucinations in SQL generarion task [4,5]. Hallucinations in LLMs refer to cases where LLM generate plausible but factually incorrect or nonsensical information [6]. Schema hallucinations and logic hallucinations are widely observed in LLM-based SQL generation [6].
>
> In this work, we present a multi-task training and a collaborative prompting framework, which significantly improved the accuracy of Text2SQL, in other words, it reduced hallucinations during the SQL generation process.
>
> Considering that the definition of hallucinations in SQL-related tasks might be unclear, we have revised the usage of SQL hallucinations in the revised submission. Please see our revised submission for details.
>
> >W4: If Schema Linking, Noise Correction, and Continuation Writing are considered important, could the authors provide the relative improvement metrics for these tasks?
>
> **Ans:** Thank you for your constructive suggestions. As suggested, we provide relative improvement indicators for three tasks. For Schema Linking (SL), we report the Recall/Precession scores of predicted related tables and columns. For Noise Correction (NC), we reports the EX scores of SQL queries refined with Noise Correction on the output SQLs of Llama3. For Continuation Writing (CW), we reports the EX scores of all SQL queries obtained by continuation writing on half of ground-truth SQL queries. The results are shown in the following table:
>
> | |SPIDER-SL|SPIDER-SL|BIRD-SL|BIRD-SL|SPIDER-NC|BIRD-NC|SPIDER-CW|BIRD-CW
> |:-:|:-:|:-:|:-:|:-:|:-:|:-:|:-:|:-:|
> | MSFT|Table-R/P|Column-R/P|Table-R/P|Column-R/P|Dev-EX|Dev-EX|Dev-EX|Dev-EX|
> |        ✓|97.38/95.71|98.59/96.98|90.87/90.22|96.13/90.89|83.4 |53.4 |91.1 |73.9|
> | |88.35/76.37|94.83/91.46|83.77/75.38|89.55/86.39|72.1 |38.1 |80.3|57.6  |
>
> As observed, we can see that MSFT has improved all three tasks, especially achieves an amazing EX score of 91.1 by conducting Continuation Writing on half of ground-truth queries. In addition, to further explore the mutual influence between each task, we provide more results in **Appendix A.5** of the revised manuscript.
>
> >W5: There are inconsistencies in writing style, such as using both "text-to-SQL" and "Text-to-SQL" interchangeably. Ensuring uniform terminology would improve the clarity and professionalism of the writing.
>
> **Ans:** Thank you for your feedback. The typos and the inconsist statements have been revised. We will re-polish our paper carefully.
>
> >Q1: The noise correction process assumes access to well-curated data and high-quality schema information, which might not be available for all databases or domains. Without rigorous data preparation, the model may struggle with hallucinations, as noise correction and schema linking effectiveness are diminished when data quality is compromised.
>
> **Ans:** In our ROUTE, the basic information used for synthetic schema linking and noise correction is extracted from the databases of two high-quality datasets SPIDER and BIRD, which are widely used in NL2SQL training/evaluation and provide complete architectural information. In addition, we filter the noisy data before synthesizing the data to ensure data quality.
>
> It is undeniable that if the data is noisy and incomplete, the final performance of multi-task collaboration will be affected, as shown in the SPIDER performance of #1 and #3 in Table 4. This encourages the construction of more reliable and complete multi-task datasets in the future to further improve ROUTE.

---

> ### Author Response · Authors · 2024-11-23
> **Official reply to Reviewer TcaB [3/3]**
>
> >Q2: In low-resource settings where high-quality SQL annotations or database schema information might be scarce, could ROUTE be enhanced by incorporating weak supervision, unsupervised learning, or semi-supervised data to fill gaps?
>
> **Ans:** Thank you for your valuable question. we think your view is valuable and feasible.
>
> There are now a large number of high quality, public Text-to-SQL datasets, either LLM-synthesized or human annotated [7,8]. We have collected a SFT dataset comprising approximately 1 million entries for Text2SQL task. Despite the large scale, there remains a prevalence of low-quality data, and scenarios involving complex queries are relatively underrepresented. Therefroe, it is valuable to explore weak supervision, unsupervised learning or semi-supervised paradigm to enrich and refine the training data. For example, a lot of apporoaches has been proposed to synthesize Text2SQL training data [9,10].
>
> However, in this paper, we primarily follow the mainstream approaches[2,9], focusing on the supervised fine-tuning paradigm, using the high-quality and well-established Text2SQL training data [11,12].
>
> In summary, exploring weak supervision, unsupervised learning or semi-supervised in low-resource scenario to achieve reliable Text2SQL is promising but beyond the scope of this work. We may explore this direction in our future work.
>
> >Q3: Given that database schemas often change over time in production, can ROUTE adapt to new tables or columns without needing extensive retraining, or would these require ongoing fine-tuning?
>
> **Ans:** Good question. Our fine-tuned LLM can be directly applied to new tables without the need for additional fine-tuning in novel scenarios.
>
> During inference, we supply the input prompt with necessary information from the test database. Consequently, our ROUTE can be seamlessly applied to cross-domain datasets beyond the original training domains (SPIDER and BIRD). This is demonstrated by our results on the SPIDER variants, as illustrated in  **Section 4.2-Table 2**.
>
> Additionally, we present the results on the Dr. spider benchmark [13], as requested by Reviewer 2daL. This benchmark includes 17 perturbation test sets designed to simulate dynamic scenarios in real-world applications. Our findings, displayed in the following table, clearly demonstrate that our ROUTE method maintains a significant advantage. The results indicate that our approach can be effectively applied to dynamic environments without compromising its performance.
>
> **Table 13 in Appendix A.5:** The performance on Dr.Spider benchmark.
>
> | |Avg.DB|Avg.NLQ|Avg.SQL|Avg.all |
> |-|-|-|-|-|
> | Methods|Pre~Post|Pre~Post|Pre~Post|Pre~Post |
> | Llama3|70.1~54.3|70.6~56.8|69.1~65.5|69.9~58.8 |
> | Llama3 + MCP|75.6~59.1|77.0~61.7|74.8~72.6|75.8~64.4 |
> | Llama3 + SFT|83.4~66.0|83.0~72.8|79.9~77.6|82.1~72.2 |
> | Llama3 + SFT + MCP|85.4~68.0|85.3~75.1|84.3~81.7|85.0~74.9 |
> | Llama3 + MSFT|83.8~66.3|82.9~72.5|80.0~77.5|82.2~72.1 |
> | **Llama3 + ROUTE**|**86.7~69.6**|**85.4~75.8**|**84.5~81.9**|**85.5~75.8** |
>
>
> ### Summary
> We have noted that your main concerns (see in weakness) about our work arise from the unclear definitions or descriptions of several key terms, such as <Question, SQL>, "SQL hallucinations," and "Overfitting in Text2SQL". We acknowledge and appreciate your feedbacks, and have revised the expressions that may lead to confusions.
>
> Considering that you have not provided significant negative feedbacks on our work in terms of motivation, technical contribution, and experimental performance, we kindly request that you re-evaluate our work based on our detailed explanations.
>
> ### Reference
> >[1] Din-sql: Decomposed in-context learning of text-to-sql with self-correction, NeureIPS, 2023.\
> >[2] Dts-sql: Decomposed text-to-sql with small large language models, arxiv preprint, 2024.\
> >[3] Benchmarking the text-to-sql capability of large language models: A comprehensive evaluation[J], arXiv preprint, 2024.\
> >[4] Before Generation, Align it! A Novel and Effective Strategy for Mitigating Hallucinations in Text-to-SQL Generation, ACL 2024\
> >[5] PURPLE: Making a Large Language Model a Better SQL Writer, ICDE 2024.\
> >[6] A Survey on Hallucination in Large Language Models: Principles, Taxonomy, Challenges, and Open Questions, ACM TOIS.\
> >[7] https://huggingface.co/datasets/philikai/200k-Text2SQL \
> >[8] https://huggingface.co/datasets/gretelai/synthetic-text-to-sql \
> >[9] Codes: Towards building open-source language models for text-to-sql, ACM SIGMOD 2024.\
> >[10] Synthesizing text-tosql data from weak and strong llms, ACL 2024.\
> >[11] Spider: A large-scale human-labeled dataset for complex and cross-domain semantic parsing and text-to-sql task, EMNLP 2018.\
> >[12] Can llm already serve as a database interface? a big bench for large-scale database grounded text-to-sqls, NeureIPS, 2023.\
> >[13] Dr.Spider: A Diagnostic Evaluation Benchmark towards Text-to-SQL Robustness, ICLR, 2023.

---

> > ### Comment · Reviewer_TcaB · 2024-11-25
> >
> > Thanks to the authors for their detailed responses, I will increase my score accordingly.

---

> > > ### Author Response · Authors · 2024-11-25
> > >
> > > Dear Reviewer TcaB,
> > >
> > > We sincerely thank you for your timely feedback and increased score.
> > >
> > > Since your current score is still below the acceptance threshold, we would appreciate any additional concerns you may have about our submission. We will do our utmost to address any remaining concerns during the rebuttal stage (about **48** hours left). Therefore, we sincerely hope you can offer further suggestions or feedbacks to help us improve the manuscript to meet the acceptance threshold.
> > >
> > > Thank you again for your feedbacks and for taking the time to consider our response. We would appreciate it if you could consider raising your score to support the acceptance of this submission, if you have no further concerns.
> > >
> > >
> > > Best,
> > > Authors

---

> > > > ### Comment · Reviewer_TcaB · 2024-11-27
> > > > **Thanks for your responses**
> > > >
> > > > Thank you to the authors for their responses and efforts. However, my remaining concern is the novelty of applying multi-task learning in the text-to-SQL field (Although the authors propose many interesting points, none of them is particularly new in this area.). Therefore, I will temporarily maintain my score.

---

> ### Author Response · Authors · 2024-11-27
> **Official reply to Reviewer TcaB's remaining concerns [1/2]**
>
> We sincerely appreciate your timely and valuable feedback. To address your concerns, we would like to further elaborate on the innovations of our ROUTE and highlight how it differs from existing methods.
>
> The valuable insights and significant contributions  of our work can be summarized as follows:
> - ROUTE is among the pioneering frameworks under the context of LLMs that explores multi-task tuning and collaborative prompting to improve Text2SQL performance.
> - We have exhaustively introduced and defined three important tasks in SQL generation, demonstarting that multi-task tuning and collaborative prompting in Schema Linking, Noise Correction and Continuation Writing significantly improve SQL generation accuracy. The additionally introduced SQL-related tasks are well integrated during both the training and inference phases.
> - We have achieved state-of-the-art performance in 7B/14B-sized LLMs on both the widely-recognized SPIDER and BIRD benchmarks, with verified generalization and transferability across various cross-domains benchmarks and LLMs.
>
> We highlight the key similarities and differences between our ROUTE and other related works as follows:
>
> >**Multi-task Supervised Fine-tuning (MSFT)**: The method most comparable to our ROUTE approach is MAC-SQL[2], which introduces multiple task agents and demonstrates the effectiveness of fine-tuning through the use of multi-agent instructions on CodeLlama-7B.
> >- **First**, the defined tasks in MSFT for ROUTE differ from those in MAC-SQL. We have introduced a new continuation writing (CW) task to further refine the challenging SQL queries. As demonstrated in **Ans-W4**, CW holds significant potential for SQL generation. On SPIDER development set, exploring CW task is able to achieve an impressive EX score of **91.1**.
> >- **Second**, in MAC-SQL, generating instruction data for SFT involves decomposing complex questions into multiple sub-questions and constructing corresponding answers. In contrast, our approach, beyond noise correction, allows for the synthesis of SFT data for various tasks using programming functions. This makes our method more practical for large-scale multi-task data synthesis for MSFT.
> >- **Third**, in terms of performance, our ROUTE is significantly outperforms MAC-SQL based on the open-source LLM of CodeLlama-7B. The detailed results are presented in the table below.
> >
> >|Methods (fine-tuning)|SPIDER-Dev-EX|BIRD-Dev-EX|
> >|-|:-:|:-:|
> >|CodeLlama-7B (SQL-Llama) + MAC-SQL|76.3|43.9|
> >|CodeLlama-7B + ROUTE|83.2|52.2|
> >
> > **SQL-Data Synthesis**: Our ROUTE involves the synthesis of SQL-related instruction-following data, which shares similarities with the recent work SENSE[3].
> >- **First**, compared to SENSE, the data synthesis pipeline of ROUTE encompasses not only Text2SQL but also multiply other SQL-related tasks. Our approach focuses on utilizing existing data to synthesize multi-task SFT data, thereby enhancing the capabilities of open-source LLMs to handle various SQL-related tasks. In contrast, SENSE mainly focused on SQL generation task, leveraging strong LLMs to increase the diversity of SQL generation training set and synthesize preference data.
> >- **Besides**, our ROUTE achieves comparable performance to SENSE on the SPIDER development set and better performance on the BIRD development set, as shown in following table.
> >
> >**Table 10 (Partially) in Appendix A.2**: The performance (EX) of different open-source LLMs.
> >|Methods (fine-tuning)|SPIDER-Dev-EX|BIRD-Dev-EX|
> >|-|:-:|:-:|
> >|CodeLlama-7B + SENSE|83.2|51.8|
> >|CodeLlama-7B + ROUTE|83.2|52.2|

---

> > ### Author Response · Authors · 2024-11-27
> > **Official reply to Reviewer TcaB's remaining concerns [2/2]**
> >
> > > **Multi-tasking Collaboration**: To exploit the potential of multi-task capabilities, we propose a multi-task collaborative prompting strategy (MCP) to improve the final SQL generation. The most similar works are DIN-SQL[1] and MAC-SQL[2], which both aim to reduce the complexity of the Text2SQL and improve the final performance via self-correction.
> > >- **First**, compared to them, our MCP efficiently integrates multiple tasks using concise prompts across all tasks, which makes it more effective in small-sized LLMs that struggle with comprehending complex instructions. As shown in the results of **Section 4.1-Table 1**, the effectiveness of MAC-SQL and DIN-SQL is constrained by the limited capacity of small-sized LLMs to comprehend complex instructions, while our MCP can achieve better and impressive performance.
> > >- **Besides**, while all three methods employ a self-correction strategy to enhance the quality of generated SQL queries, our MCP introduces a novel continuation writing task specifically designed to refine challenging SQL queries and improve the performance significantly.
> > >
> > >**Table 1 (Partially) in Section 4.2**: Performance comparison on SPIDER and BIRD benchmarks.
> > >||SPIDER|SPIDER|SPIDER|BIRD|BIRD|
> > >|-|-|-|-|-|-|
> > >|Methods/LLMs|Dev-EX|Dev-TS|Test-EX|Dev-EX|Dev-VES|
> > >|Llama3-8B|69.3|58.4|69.1|32.1|31.6|
> > >|DIN-SQL + Llama3-8B|48.7|39.3|47.4|20.4|24.6|
> > >|MAC-SQL + Llama3-8B|64.3|52.8|65.2|40.7|40.8|
> > >|**MCP + Llama3-8B**|**75.0**|**63.4**|**72.0**|**42.7**|**44.8**|
> > >|Qwen2.5-7B|72.5|64.0|75.9|41.1|42.0|
> > >|DIN-SQL + Qwen2.5-7B|72.1|61.2|71.1|30.1|32.4|
> > >|MAC-SQL + Qwen2.5-7B|71.7|61.9|72.9|46.7|49.8|
> > >|**MCP + Qwen2.5-7B**|**78.3**|**67.2**|**78.7**|**49.7**|**52.8**|
> >
> > Considering the comprehensive nature of our work, which encompasses data synthesis, supervised fine-tuning, and multi-task collaborative prompting, it is inevitable that there are some similarities with existing work. Nevertheless, we have offered numerous insights into the Text2SQL task and achieved promising results, which we believe are significant contributions to the Text2SQL community.
> >
> > We thank you again for your timely feedback and valuable comments. We have added the above discussion in **Appendix A.8** to further clarify our innovations and contributions.
> >
> > ### Reference
> > > [1] Din-sql: Decomposed in-context learning of text-to-sql with self-correction, NeureIPS, 2023.\
> > > [2] MAC-SQL: A Multi-Agent Collaborative Framework for Text-to-SQL, arxiv preprint, 2024.\
> > > [3] Synthesizing text-tosql data from weak and strong llms, ACL 2024.

---

### Official Review · Reviewer_cCGH · 2024-11-01

**Soundness:** 4
**Presentation:** 3
**Contribution:** 3
**Rating:** 8
**Confidence:** 5

**Summary:**

The paper introduces “ROUTE,” a novel approach to (1) finetune open-source large language models (LLMs) for Text-to-SQL through multi-task supervised fine-tuning (MSFT) and (2) leverage multitask collaboration prompting (MCP) for SQL generation during inference. The MSFT tasks include Text-to-SQL, Schema Linking, Noise Correction, and Continuation Writing. The proposed method aims to reduce hallucinations and enhance Text-to-SQL robustness, demonstrated by improved performance on two benchmarks: Spider and BIRD.

**Strengths:**

1. The paper introduces a multitask learning approach that leverages several text-to-SQL related tasks. Noise Correction is designed to assess whether the execution result of a SQL query correctly answers the question, reducing hallucinations when paired with multi-turn generation.
2. ROUTE demonstrates competitive accuracy, outperforming some closed-source methods on benchmarks, thus showcasing the effectiveness of multitask training over single-task fine-tuning.
3. The authors provide comprehensive experiments using multiple LLMs as base models, demonstrating that ROUTE is generalizable across various LLMs.

**Weaknesses:**

1. The paper lacks an ablation study on the contribution of each task in MSFT. For instance, the loss from continuation writing is likely already included in text-to-SQL learning after the first token of the SQL prediction. It is unclear how each task directly benefits SQL generation and other inference components.
2. Although Noise Correction helps improve performance, it relies on execution results within the model, which may be difficult to apply to queries with large outputs, such as selecting an entire column.
3. While ROUTE demonstrates strong performance on Spider variants compared to baselines, it remains unclear whether these gains are due to improved robustness or general text-to-SQL performance. It would also be valuable to understand how each component contributes to robustness specifically. Dr. Spider [1] is a more comprehensive perturbation dataset with relative robustness evaluation, which could be useful for evaluating ROUTE’s improvement more clearly.
[1] https://arxiv.org/pdf/2301.08881

**Questions:**

For Noise Correction, is it able to handle a large table as the execution result?

---

> ### Author Response · Authors · 2024-11-23
> **Official reply to Reviewer cCGH [1/2]**
>
> We thank Reviewer cCGH for the positive feedback and constructive suggestions that contribute to improving this paper. Below, we address your comments point by point.
>
> >W1: The paper lacks an ablation study on the contribution of each task in MSFT. For instance, the loss from continuation writing is likely already included in text-to-SQL learning after the first token of the SQL prediction. It is unclear how each task directly benefits SQL generation and other inference components.
>
> **Ans:** Thank you for your constructive suggestions. We conducted additional experiments to explore the impact of single-task SFT and MSFT on each task.
> For Text-to-SQL (TS), we report zero-shot EX results on the SPIDER and BIRD development sets. For Schema Linking (SL), we report the Recall/Precession scores of predicted related tables and columns. For Noise Correction (NC), we reports the EX scores of SQL queries refined with Noise Correction on the output SQLs of Llama3. For Continuation Writing (CW), we reports the EX scores of all SQL queries obtained by continuation writing on half of ground-truth SQL queries. The specific experimental results are shown in following table, where '--' means that LLMs cannot obtain the output in the expected format due to overfitting.
>
> **Table 15 in Appendix A.6:** The SFT impact of all tasks on each other.
>
> || |SPIDER-TS|BIRD-TS|SPIDER-SL|SPIDER-SL|BIRD-SL|BIRD-SL|SPIDER-NC|BIRD-NC|SPIDER-CW|BIRD-CW |
> |-|:-|:-:|:-:|:-:|:-:|:-:|:-:|:-:|:-:|:-:|:-:|
> |No.| Settings|Dev-EX|Dev-EX|Table-R/P|Column-R/P|Table-R/P|Column-R/P|Dev-EX|Dev-EX|Dev-EX|Dev-EX |
> | #1|Llama3 with MSFT|83.6 |53.6 |97.38/95.71|98.59/96.98|90.87/90.22|96.13/90.89|83.4 |53.4 |91.1 |73.9|
> | #2|SFT with TS|83.1 |52.9 |--|--|--|--|--|--|85.6 |69.2|
> | #3|SFT with SL|--|--|95.55/92.69|98.91/95.29|87.84/85.11|94.93/89.51 |--|--|--|-- |
> | #4|SFT with NC|0.1 |8.7 |--|--|--|--|78.9 |49.3 |48.6 |38.6|
> | #5|SFT with CW|68.1 |39.0 |--|--|--|--|--|--|89.8 |70.1|
> | #6|Llama3 w/o SFT|69.3 |32.1 |88.35/76.37|94.83/91.46|83.77/75.38|89.55/86.39|72.1 |38.1 |80.3 |57.6|
>
> From the results, we can see that although single-task SFT can improve the ability to process the corresponding task, it can easily cause overfitting on instructions and lead to the degradation of the ability to process other tasks (e.g., **No.#2,#3,#4,#5**), which is not conducive to multi-task collaboration. On the contrary, our MSFT (**No.#1**) can boost the ability of each task while reducing the risk of overfitting, thus improving the feasibility of multi-tasking collaboration.
>
>
> >W2: Although Noise Correction helps improve performance, it relies on execution results within the model, which may be difficult to apply to queries with large outputs, such as selecting an entire column.
>
> **Ans:** Thank you for your careful review. We have to clarify that our noise correction does not need to put large outputs into Promt, but the status results of SQL query execution, such as error exception information. To this end, our noise correction is able to handle such SQL queries with large output. To prevent readers from being confused, we have clarified it again in **lines 286~288** of revised version. Thank you again for your careful comments.

---

> ### Author Response · Authors · 2024-11-23
> **Official reply to Reviewer cCGH [2/2]**
>
> >W3: While ROUTE demonstrates strong performance on Spider variants compared to baselines, it remains unclear whether these gains are due to improved robustness or general text-to-SQL performance. It would also be valuable to understand how each component contributes to robustness specifically. Dr.Spider [1] is a more comprehensive perturbation dataset with relative robustness evaluation, which could be useful for evaluating ROUTE’s improvement more clearly. [https://arxiv.org/pdf/2301.08881](https://arxiv.org/pdf/2301.08881)
>
> **Ans:** Thank you for your constructive suggestions. As suggested, according to the suggestion, we evaluated on Dr.Spider to have a clearer and more comprehensive understanding of the advantages of our ROUTE. Dr.Spider includes 17 perturbation variants that can comprehensively measure the effectiveness and robustness. The specific experimental results are shown in the following tables, including the study on MSFT and on each component. From the results, its conclusions are consistent with those those on SPIDER and BIRD and each component brought performance improvement, which shows that each task has an indispensable contribution to the performance. This further verifies the advantages and robustness of ROUTE. Thank you again for your constructive suggestions. We have added the corresponding experimental results in **Appendix A.5** of the revised manuscript.
>
> **Table 13 in Appendix A.5:** The performance on Dr.Spider benchmark.
>
> |  | Avg.DB | Avg.NLQ | Avg.SQL | Avg.all |
> |---|---|---|---|---|
> | Methods | Pre~Post | Pre~Post | Pre~Post | Pre~Post |
> | Llama3 | 70.1~54.3 | 70.6~56.8 | 69.1~65.5 | 69.9~58.8 |
> | Llama3 + MCP | 75.6~59.1 | 77.0~61.7 | 74.8~72.6 | 75.8~64.4 |
> | Llama3 + SFT | 83.4~66.0 | 83.0~72.8 | 79.9~77.6 | 82.1~72.2 |
> | Llama3 + SFT + MCP | 85.4~68.0 | 85.3~75.1 | 84.3~81.7 | 85.0~74.9 |
> | Llama3 + MSFT | 83.8~66.3 | 82.9~72.5 | 80.0~77.5 | 82.2~72.1 |
> | **Llama3 + ROUTE** | **86.7~69.6** | **85.4~75.8** | **84.5~81.9** | **85.5~75.8** |
>
> **Table 14 in Appendix A.5:** The ablation results (EX) on Dr.Spider.
>
> |  |  |  |   | Avg.DB | Avg.NLQ | Avg.SQL | Avg.all |
> |---|---|---|---|---|---|---|---|
> | No. | SL | NC | CW | Pre~Post | Pre~Post | Pre~Post | Pre~Post |
> | #1 |        ✓ |        ✓ |        ✓ | **86.7~69.6** | **85.4~75.8** | **84.5~81.9** | **85.5~75.8** |
> | #2 |        ✓ |  |  | 86.3~67.9 | 84.6~75.4 | 84.4~82.1 | 85.1~75.1 |
> | #3 |  |        ✓ |  | 84.4~67.7 | 83.7~73.7 | 80.3~78.7 | 82.8~73.3 |
> | #4 |  |  |        ✓ | 84.0~66.6 | 83.0~73.0 | 80.1~78.0 | 82.4~72.6 |
> | #5 |  |  |  | 83.8~66.6 | 82.9~72.5 | 80.0~77.5 | 82.2~72.1 |
>
>
> >Q: For Noise Correction, is it able to handle a large table as the execution result?
>
> **Ans:** Thank you for your valuable question. We have to cliam thatThe input of our noise correction is <Schema, Question, SQL, Execution Information>, where the execution information only refers to the exception information of the SQL executor. If the execution passes, there is no information. Therefore, our noise correction is able to handle SQL involving the large tables instead of using the results of the entire table as a prompt. To prevent readers from being confused, we have clarified it again in **lines 286~288** of revised version. Thank you again for your valuable comments.
>
> ### Reference
> >[1] Dr.Spider: A Diagnostic Evaluation Benchmark towards Text-to-SQL Robustness, ICLR, 2023.

---

> > ### Comment · Reviewer_cCGH · 2024-11-27
> >
> > Thank you to the authors for providing additional experiments. My concerns have been mostly addressed, and I have increased my scores to reflect this. If time allows, I would suggest conducting an ablation study by removing one task from the MSFT instead of using only one task in SFT for rows #2, #3, #4, and #5 in Table 15. I believe it is useful to have this experiment added in either the rebuttal or a later revision of the paper.

---

> ### Author Response · Authors · 2024-11-30
>
> Dear Reviwer cCGH,
>
> Thank you for raising the score and valuable suggestions.
>
> We have supplemented the ablation experimental results of removing each task from MSFT. The results are presented in the following table.
>
> | ||SPIDER-TS|BIRD-TS|SPIDER-SL|SPIDER-SL|BIRD-SL|BIRD-SL|SPIDER-NC|SPIDER-NC|SPIDER-CW|SPIDER-CW |
> |:-:|:-|:-:|:-:|:-:|:-:|:-:|:-:|:-:|:-:|:-:|:-:|
> | No.|Settings|Dev-EX|Dev-EX|Table-R/P|Column-R/P|Table-R/P|Column-R/P|Dev-EX|Dev-EX|Dev-EX|Dev-EX |
> | #1|Llama3 with MSFT|83.6 |53.6 |97.38/95.71|98.59/96.98|90.87/90.22|96.13/90.89|83.4 |53.4 |91.1 |73.9|
> | #2|MSFT w/o TS|0.1 |16.2 |96.58/93.94|98.40/96.32|90.79/88.26|95.95/90.34|77.4 |45.5 |86.5 |69.6|
> | #3|MSFT w/o SL|81.8 |50.9 |--|--|--|--|76.3 |47.4 |91.3 |73.5|
> | #4|MSFT w/o NC|82.8 |51.0 |96.52/94.25|99.00/96.59|90.41/88.85|96.09/90.75|--|--|91.7 |73.4|
> | #5|MSFT w/o CW|81.2 |50.3 |96.51/93.97|98.65/96.39|90.59/88.12|96.05/90.64 |79.4 |49.0 |81.2 |56.7|
> | #6|SFT with TS|83.1 |52.9 |--|--|--|--|--|--|85.6 |69.2|
> | #7|SFT with SL|--|--|95.55/92.69|98.91/95.29|87.84/85.11|94.93/89.51 |--|--|--|-- |
> | #8|SFT with NC|0.1 |8.7 |--|--|--|--|78.9 |49.3 |48.6 |38.6|
> | #9|SFT with CW|68.1 |39.0 |--|--|--|--|--|--|89.8 |70.1|
> | #10|Llama3 w/o SFT|69.3 |32.1 |88.35/76.37|94.83/91.46|83.77/75.38|89.55/86.39|72.1 |38.1 |80.3 |57.6|
>
>
> The results suggest that tasks not included in MSFT demonstrate lower performance due to the overfitting of LLMs to other tasks. This highlights the importance of considering SFT across multiple tasks to prevent performance degradation of an LLM when handling additional tasks.
>
> Furthermore, the results indicate that although the performance of the full MSFT on certain tasks, such as SL and CW, is somewhat inferior to that of single or triple-task SFT, the full MSFT demonstrates significant performance improvements across each task and exhibits better stability.
>
> While we can not upload the revised manuscript at this time, we will update the ablation results and corresponding discussion in the next version following the rebuttal stage.
>
> Thank you again for your valuable feedback and the positive score.
>
> Best,
> Authors

---

> > ### Comment · Reviewer_cCGH · 2024-12-03
> >
> > Thank you for further addressing my questions.

---

### Official Review · Reviewer_2daL · 2024-11-02

**Soundness:** 3
**Presentation:** 3
**Contribution:** 3
**Rating:** 6
**Confidence:** 4

**Summary:**

This paper addresses the limitations of current Text-to-SQL approaches that rely heavily on in-context learning using closed-source LLMs, such as GPT-4, which can cause privacy issues. To overcome these issues, the authors propose ROUTE, a comprehensive solution to enhance open-source LLMs' Text2SQL capabilities. ROUTE utilizes a multitask supervised fine-tuning approach incorporating tasks like text-to-SQL, schema linking, noise correction, and continuation writing to broaden the model's SQL generation skills and reduce the risk of overfitting. Additionally, a Multitask Collaboration Prompting (MCP) strategy is employed during inference to decompose the SQL generation process into simpler sub-tasks, reducing hallucinations and improving performance.

**Strengths:**

1) The proposed method significantly improves the performance of open-source LLMs and outperforms all existing methods trained on open-source LLMs.

2) The proposed MCP approach not only enhances the performance of models trained with MSFT but also improves other models.

3) The novel MSFT method substantially boosts model performance compared to standard SFT.

**Weaknesses:**

1) Although this paper focuses more on open-source LLMs, some recent approaches, such as CHASE-SQL, Distillery, and CHESS, are not included as benchmarks in their experiments.

2) The proposed approach is a multi-step pipeline that can be prone to error propagation. To better understand the performance of the schema linking module and ensure it is not introducing errors into the pipeline, it would be beneficial to report the precision and recall of the schema linking module, as done in CHESS and DTS-SQL.

3) The performance gap with the close-source LLMs is still large, roughly 13% on BIRD development set, which makes the applicability of this approach limited to the scenarios where privacy and local LLMs is essential.

**Questions:**

1). For the open-source LLMs and super large databases such as some of the databases in BIRD benchmark, how these large schema are fitted into the prompt of the open-source models?

---

> ### Author Response · Authors · 2024-11-23
> **Official reply to Reviewer 2daL [1/2]**
>
> We thank Reviewer 2daL for the positive feedback and constructive suggestions that contribute to improving this paper. Below, we provide one-on-one responses to your comments.
>
> >W1: Although this paper focuses more on open-source LLMs, some recent approaches, such as CHASE-SQL, Distillery, and CHESS, are not included as benchmarks in their experiments.
>
> **Ans:** Thank you for your valuable suggestion, we have included several recent methods in the revised version to ensure a comprehensive comparison and review of related work. Below is a brief summary, please refer to **Appendix A.3** in our revised submission for the details.
>
> **Table 11 in Appendix A.3:** The comparisons with recent methods on SPIDER and BIRD.
>
> | Methods | SPIDER-Dev-EX | BIRD-Dev-EX | BIRD-Dev-VES |
> |---|:---:|:---:|:---:|
> | CHASE-SQL + Gemini 1.5 | 87.6  | 73.1 | 73.0  |
> | Distillery + GPT-4o | - | 67.2  | 72.9  |
> | CHESS + proprietary (GPT-4) | 87.2  | 65.0  | 65.4  |
> | ROUTE + Qwen2.5-14B | 87.3  | 60.8  | 65.2  |
>
> From the results, we observed that:
> - On SPIDER, our ROUTE-14B achieves similar performance as CHESS based on Gimini or GPt-4/4o. This suggests that our ROUTE is an exceptional choice in both conventional and privatized Text2SQL scenarios.
>
> - On BIRD, our ROUTE fall behind CHESS + proprietary by 5 point and CHASS-SQL + Gemini 1.5 by 12 point. We believe this is due to the complexity of the BIRD database, which contains numerous tables and columns in a single database, resulting in an extensive input context. It is widely recognized that smaller-sized LLMs (7B/14B) have relative limitations in reasoning capabilities and managing lengthy texts.
>
> >W2: The proposed approach is a multi-step pipeline that can be prone to error propagation. To better understand the performance of the schema linking module and ensure it is not introducing errors into the pipeline, it would be beneficial to report the precision and recall of the schema linking module, as done in CHESS and DTS-SQL.
>
> **Ans:** Thank you for your constructive feedback. We have reported the Recall and Precession results in **Appendix A.4** of our revised submission. We also present the results as follows for convenience, which demonstartes several important observations:
> - After MSFT, the schema linking capability is significantly improved, especially in terms of precision scores.
> - A higher Recall score generally leads to improved EX performance due to minor information loss.
> - While simplifying the database schema is necessary, ensuring its completeness is more crucial for achieving enhanced performance.
>
> **Table 12 in Appendix A.4:** The performance of schema linking.
>
> | 　||SPIDER|Table|Table|Column|Column|BIRD|Table|Table|Column|Column |
> |-|-|:-:|:-:|:-:|:-:|:-:|:-:|:-:|:-:|:-:|:-:|
> | $\text{SL}_{\sigma_s}$|$\text{SL}_{\sigma_t}$|Dev-EX|Recall|Precession|Recall|Precession|Dev-EX|Recall|Precession|Recall|Precession |
> | ✓ |✓ |85.80 |**98.75** |94.67 |**99.24** |96.36 |56.00 |**95.21** |88.50 |**96.95**|89.93|
> | ✓ ||84.10 |97.38 |95.71 |98.59 |96.98 |52.70 |90.87 |90.22 |96.13 |90.89|
> | |✓ |85.00 |97.01 |97.26 |98.21 |97.99 |54.50 |91.60 |93.14 |94.15 |94.11|
> | ||83.30 |100.00 |18.27 |100.00 |40.05 |53.10 |100.00 |12.23 |100.00 |31.64|
> | ✓ |✓ |73.30 |**97.43** |74.99 |**98.61** |90.33 |36.80 |**93.65**|73.57 |**95.78** |84.36|
> | ✓ ||64.50 |88.35 |76.37 |94.83 |91.46 |30.40 |83.77|75.38 |89.55 |86.39|
> | |✓ |73.10 |94.24 |91.60 |97.12 |95.30 |35.40 |82.15 |89.01 |88.32 |91.63|
> | ||69.30 |100.00 |18.27 |100.00 |40.05 |32.10 |100.00 |12.23 |100.00 |31.64|
>
>
> Note that the first four rows are the results of post-MSFT LLMs, and the last four rows are the that of the original LLMs.

---

> > ### Comment · Reviewer_2daL · 2024-11-27
> > **Thanks for the efforts**
> >
> > Thank you so much for your efforts and addressing some of my concerns.
> >
> > > On SPIDER, our ROUTE-14B achieves similar performance as CHESS based on Gimini or GPt-4/4o
> >
> > Although the performance is on par with the CHESS and CHASE approaches on the Spider dataset, it is important to note that both of these approaches evaluated their pipelines on the Spider dataset without any fine-tuning or prompt optimization. This was done to demonstrate the generalizability of their methods. In contrast, ROUTE is a fine-tuned model, making this comparison somewhat less equitable.
> >
> > > We have reported the Recall and Precession results
> >
> > Thank you so much for sharing these insightful results. As a suggestion, I believe it would be valuable to include this in the main paper.

---

> ### Author Response · Authors · 2024-11-23
> **Official reply to Reviewer 2daL [2/2]**
>
> >W3: The performance gap with the close-source LLMs is still large, roughly 13% on BIRD development set, which makes the applicability of this approach limited to the scenarios where privacy and local LLMs is essential.
>
> **Ans:** Thank you for your valuable comments. It is undeniable that our solution still has a certain gap with the solution using closed -source LLMs, but the solution using open-source models has unique advantages in privacy and local LLMs. It can be customized according to business data sets. Also, the smaller model reasoning speed is relatively fast and can better meet the needs of multi-step reasoning. In addition, our ROUTE is a general solution that is better than existing SFT methods (e.g., DTS-SQL and SENSE). From our experiments, we can see that it can be used in almost all kinds of open-source LLMs without paying too much cost for a considerable performance improvement. In addition, we can see that our method achieves **87.3** EX score on SPIDER development set which is close to the results of most closed-source methods. This shows that our ROUTE can be a substitute for applications in some closed-source scenarios. We believe ROUTE can contribute to the community of Text2SQL in the future.
>
> >Q1: For the open-source LLMs and super large databases such as some of the databases in BIRD benchmark, how these large schema are fitted into the prompt of the open-source models?
>
> **Ans:** Thank you for your valuable review. As the prompt template shown in **Appendix A.11**, we extracted the key information about the database to incorporate into the prompt, e.g., the column names and some example rows. In addition, It is worth noting that the context length of popular LLMs, such as LLama3 and Qwen2.5, can effectively meet the requirements of BIRD, with context lengths ranging from 8K to 32K.
>
> When dealing with a very large database and complex prompts with extensive sequence lengths, we can utilize the 128K version of open-source LLMs supported by YaRN [4]. Besides, we can also introduce special techniques to simplify the input prompts, as demonstrated by CODES [5].
>
> However, it is undeniable that the current solutions still have limitations when dealing with extremely large databases. This opens up a new avenue for future research on ROUTE. Thank you again for your valuable comments.
>
>
> ### Reference
> >[1] CHASE-SQL: Multi-Path Reasoning and Preference Optimized Candidate Selection in Text-to-SQL, arxiv preprint, 2024.\
> >[2] The Death of Schema Linking? Text-to-SQL in the Age of Well-Reasoned Language Models, arxiv preprint, 2024.\
> >[3] Chess: Contextual harnessing for efficient sql synthesis, arxiv preprint, 2024.\
> >[4] YaRN: Efficient Context Window Extension of Large Language Models, ICLR, 2024.\
> >[5] Codes: Towards building open-source language models for text-to-sql, ACM SIGMOD 2024.

---

> ### Author Response · Authors · 2024-11-27
>
> Dear Reviewer 2daL,
>
> Thank you for your timely feedback and constructive suggestions.
>
> Our ROUTE focuses on the supervised fine-tuning paradigm for SQL generation using small-sized LLMs. This inevitably leads to bias and an unfair comparison with recent prompting-based methods that utilize GPT-4/4o (e.g., CHESS and CHASE). However, in our experiments, our baselines basically include recent advanced SFT-based methods such as CODES, SENSE, and DTS-SQL, to verify the effectiveness and superiority of ROUTE. We have also verified the generalization of our approach on multiple open-source LLMs and benchmarks.
>
> In addition, due to the space limitation, we have provided hints about the Recall and Precession results in the revised manuscript: **lines 443~444** (Section 4.3-Study on Enhanced Schema Linking) to draw readers’ attention to these insightful results.
>
> We thank you again for your timely feedback and valuable comments. Your suggestions have been of great help in improving the quality of our manuscript. If you have any further questions or suggestions, feel free to ask.
>
> Best,
> Authors

---

### Author Response · Authors · 2024-11-23
**Meta Reply**

We sincerely thank all reviewers for spending their time to carefully review our paper and providing constructive suggestions, which helped to further improve our manuscript. Here, we provide a comprehensive overview of the reviewers' feedback and outline the corresponding modifications we have made.

### The merits of our submission
1. The significant performance improvements and superior performance on open-source LLMs. (Reviewer 2daL, Ej7x)
2. The experiments are comprehensive and the effectiveness of the proposed method is verfied. (Reviewer cCGH, TcaB, Ej7x)
3. The paper is easy-to-follow. (Reviewer Ej7x)

### Important suggestions and concerns
1. A more comprehensive evaluation on the Dr.Spider benchmark.
2. Accuracy evaluation on the task of schema linking.
3. The impact of each SQL-related task and the relative improvement.
4. The clarification on ‘hallucinations’.
5. The clarification on differences with existing methods.

### Our Revisions
According to the reviewers' constructive and insightful feedbacks, we have carefully revised our paper, and provided comprehensive experiments, analysis in the appendix.

- In main text, we double-checked and modified some descriptions based on the modification suggestions and highlighted them in blue.
- In Appendix A.3, we provide more comparison results with recent works.
- In Appendix A.4, we evaluate the performance of our schema linking module.
- In Appendix A.5, we provide the ablation results on Dr.Spider benchmark to further verify the effectiveness of our method.
- In Appendix A.6, we explore the impact of single-task SFT to further understand the interplay of multiple tasks.
- In Appendix A.7, we further clarified some details in the step of noisy correspondence filtering.
- In Appendix A.8, we further discuss the innovations of our ROUTE and how it differs from recent related methods.

Finally, we sincerely thank all reviewers and ACs for their efforts.

---

### Meta-Review · Area_Chair_Q4zL · 2024-12-21

**Metareview:**

**Strengths:**

* The proposed method “significantly improves the performance of open-source LLMs and outperforms all existing methods trained on open-source LLMs” (2daL, also noted by cCGH), and “achieves comparable performance to GPT-4” (Ej7x).

* Results are strong across multiple well-known benchmarks, including Spider, Bird, perturbed variants of Spider, as well as Dr. Spider, “indicating the robustness of ROUTE” (Ej7x), while “demonstrating its effectiveness in real-world applications” (TcaB)

* Evaluation is comprehensive “using multiple LLMs as base models” (cCGH) and “an extensive list of baselines” (Ej7x).
The proposed method works in both fine-tuning and prompting regime (2daL), and makes “LLM(s) more versatile and capable of handling complex SQL generation scenarios.” (TcaB)

* “The paper is easy-to-follow, and the writing is mostly clear” (Ej7x), with some clarity issues on difference with existing methods and definition of certain terms (see weaknesses).

**Weaknesses:**

The following weaknesses are addressed during the rebuttal phase:

* Comparison with more recent prompting-based approaches (2daL, e.g., CHASE-SQL, Distillery, and CHESS)

* More careful ablation on different components of the pipeline to better understand their individual contributions to final performance (2daL, cCGH, Ej7x), while reporting quantitative evaluation metrics on the performance of individual components (e.g., schema linking, 2daL, TcaB).

* As noted by TcaB, there are a few places in the submission where the technical presentation is not clear, such as how the proposed method addresses overfitting and hallucinations in generated SQL queries.

The single remaining issue yet to be addressed after the rebuttal phase is the novelty of the method: as noted by TcaB and Ej7x, the proposed method is a combination of many existing approaches in text-to-SQL. Therefore, “it is not very clear what kind of novel contribution this paper is making” (Ej7x).  “Although the authors propose many interesting points, none of them is particularly new in this area” (TcaB).

While this submission may not present a truly novel idea, I believe this submission makes valuable contributions through strong empirical results. The fact that the paper focuses on improving the performance of open-source models via both prompting and fine-tuning is also a plus for industry practitioners to deploy similar approaches in practical scenarios. Therefore, given the rating and the strong empirical results, the recommendation is **Accept**.

**Further suggestions on revision:**

Besides existing feedback from reviewers, please focus more carefully on explaining the relations with many existing works flagged by TcaB and Ej7x. I also agree with Ej7x that some components in the framework are “rebranded under new terms”. Please avoid inventing new terms since this would cause confusion. Instead, please use well-known technical terms to refer to certain parts of your model. For example, please consider renaming “noise correction” to “self-debugging” or “program repair”.

**Additional Comments On Reviewer Discussion:**

Please see the above meta review.

---

### Decision · Program_Chairs · 2025-01-22

Accept (Poster)